# Linear Convergence and Implicit Regularization of Generalized Mirror Descent with Time-Dependent Mirrors

## Abstract

The following questions are fundamental to understanding the properties of over-parameterization in modern machine learning: (1) Under what conditions and at what rate does training converge to a global minimum? (2) What form of implicit regularization occurs through training? While significant progress has been made in answering both of these questions for gradient descent, they have yet to be answered more completely for general optimization methods. In this work, we establish sufficient conditions for linear convergence and obtain approximate implicit regularization results for *generalized mirror descent* (GMD), a generalization of mirror descent with a possibly time-dependent mirror. GMD subsumes popular first order optimization methods including gradient descent, mirror descent, and preconditioned gradient descent methods such as Adagrad. By using the Polyak-Lojasiewicz inequality, we first present a simple analysis under which non-stochastic GMD converges linearly to a global minimum. We then present a novel, Taylor-series based analysis to establish sufficient conditions for linear convergence of stochastic GMD. As a corollary, our result establishes sufficient conditions and provides learning rates for linear convergence of stochastic mirror descent and Adagrad. Lastly, we obtain approximate implicit regularization results for GMD by proving that GMD converges to an interpolating solution that is approximately the closest interpolating solution to the initialization in $\ell_2$-norm in the dual space.

## 1 Introduction

Recent work has established the optimization and generalization benefits of over-parameterization in machine learning (Belkin et al., 2019; Liu et al., 2020; Zhang et al., 2017). In particular, several works including Vaswani et al. (2019); Du et al. (2018); Liu et al. (2020); Li & Liang (2018) have demonstrated that over-parameterized models converge to a global minimum when trained using stochastic gradient descent and that such convergence can occur at a linear rate. Independently, other work, such as Gunasekar et al. (2018), have characterized *implicit regularization* of over-parameterized models, i.e., the properties of the solution selected by a given optimization method, without proving convergence.

Recently, Azizan & Hassibi (2019); Azizan et al. (2019) simultaneously proved convergence and analyzed approximate implicit regularization for mirror descent (Beck & Teboulle, 2003; Nemirovsky & Yudin, 1983). In particular, by using the fundamental identity of stochastic mirror descent (SMD), they proved that SMD converges to an interpolating solution that is approximately the closest one to the initialization in Bregman divergence. However, these works do not provide a rate of convergence for SMD and assume that there exists an interpolating solution within $\epsilon$ in Bregman divergence from the initialization. In this work, we provide sufficient conditions for *linear* convergence and obtain approximate implicit regularization results for *generalized mirror descent* (GMD), an extension of mirror descent that introduces (1) a potential-free update rule and (2) a time-dependent mirror; namely, GMD with invertible $\phi : \mathbb{R}^d \to \mathbb{R}^d$ and learning rate $\eta$ is used to minimize a real valued loss function, $f$, according to the update rule:

$$\phi^{(t)}(w^{(t+1)}) = \phi^{(t)}(w^{(t)}) - \eta \nabla f(w^{(t)}). \tag{1}$$

We discuss the stochastic version of GMD (SGMD) in Section 3. GMD generalizes both mirror descent and preconditioning methods. Namely, if for all $t$, $\phi^{(t)} = \nabla \psi$ for some strictly convex function $\psi$, then GMD corresponds to mirror descent with potential $\psi$; if $\phi^{(t)} = G^{(t)}$ for some invertible matrix $G^{(t)} \in \mathbb{R}^{d \times d}$, then the update rule in equation (1) reduces to

$$w^{(t+1)} = w^{(t)} - \eta G^{(t)^{-1}} \nabla f(w^{(t)})$$

and hence represents applying a pre-conditioner to gradient updates. The following is a summary of our results:

1. We provide a simple proof for linear convergence of GMD under the Polyak-Lojasiewicz inequality (Theorem 1).

2. We provide sufficient conditions under which SGMD converges linearly under an adaptive learning rate (Theorems 2 and 3)[1].

3. As corollaries to Theorems 1 and 3, in Section 5 we provide sufficient conditions for linear convergence of stochastic mirror descent as well as stochastic preconditioner methods such as Adagrad (Duchi et al., 2011).

4. We prove the existence of an interpolating solution and linear convergence of GMD to this solution for non-negative loss functions that locally satisfy the PL* inequality (Liu et al., 2020). This result (Theorem 4) provides approximate implicit regularization results for GMD: GMD converges linearly to an interpolating solution that is approximately the closest interpolating solution to the initialization in $\ell_2$ norm in the dual space induced by $\phi^{(t)}$.

## 2 RELATED WORK

Recent work (Azizan et al., 2019) established convergence of stochastic mirror descent (SMD) for nonlinear optimization problems. It characterized the implicit bias of mirror descent by demonstrating that SMD converges to a global minimum that is within epsilon of the closest interpolating solution in Bregman divergence. The analysis in Azizan et al. (2019) relies on the fundamental identity of SMD and does not provide explicit learning rates or establish a rate of convergence for SMD in the nonlinear setting. The work in Azizan & Hassibi (2019) provided explicit learning rates for the convergence of SMD in the linear setting under strongly convex potential, again without a rate of convergence. While these works established convergence of SMD, prior work by Gunasekar et al. (2018) analyzed the implicit bias of SMD without proving convergence.

A potential-based version of generalized mirror descent with time-varying regularizes was presented for online problems in Orabona et al. (2015). That work is primarily concerned with establishing regret bounds for the online learning setting, which differs from our setting of minimizing a loss function given a set of known data points. A potential-free formulation of GMD for the flow was presented in Gunasekar et al. (2020).

The Polyak-Lojasiewicz (PL) inequality (Lojasiewicz, 1963; Polyak, 1963) serves as a simple condition for linear convergence in non-convex optimization problems and is satisfied in a number of settings including over-parameterized neural networks (Liu et al., 2020). Work by Karimi et al. (2016) demonstrated linear convergence of a number of descent methods (including gradient descent) under the PL inequality. Similarly, Vaswani et al. (2019) proved linear convergence of stochastic gradient descent (SGD) under the PL inequality and the strong growth condition (SGC), and Bassily et al. (2018) established the same rate for SGD under just the PL inequality. Soltanolkotabi et al. (2019) also used the PL inequality to establish a local linear convergence result for gradient descent on 1 hiddden layer over-parameterized neural networks.

Recently, Xie et al. (2020) established linear convergence for a norm version of Adagrad (Adagrad-Norm) using the PL inequality, while Wu et al. (2019) established linear convergence for Adagrad-Norm in the particular setting of over-parameterized neural networks with one hidden layer. An alternate analysis for Adagrad-Norm for smooth, non-convex functions was presented in Ward et al. (2019), resulting in a sub-linear convergence rate.

---

[1]We also provide a fixed learning rate for monotonically decreasing gradients $\nabla f(w^{(t)})$.

Instead of focusing on a specific method, the goal of this work is to establish sufficient conditions for linear convergence by applying the PL inequality to a more general setting (SGMD). We arrive at linear convergence for specific methods such as mirror descent and preconditioned gradient descent methods as corollaries. Moreover, our local convergence results provide an intuitive formulation of approximate implicit regularization for GMD and thus mirror descent. Namely, instead of resorting to Bregman divergence, we prove that GMD converges to an interpolating solution that is approximately the closest interpolating solution to the initialization in $\ell_2$ norm in the dual space induced by $\phi^{(t)}$.

## 3    ALGORITHM DESCRIPTION AND PRELIMINARIES

We begin with a formal description of SGMD. Let $f_i : \mathbb{R}^d \to \mathbb{R}$ denote real-valued, differentiable loss functions and let $f(x) = \frac{1}{n} \sum_{i=1}^n f_i(x)$. In addition, let $\phi^{(t)} : \mathbb{R}^d \to \mathbb{R}^d$ be an invertible function for all non-negative integers $t$. We solve the optimization problem

$$\arg \min_{x \in \mathbb{R}^d} f(x)$$

using **stochastic generalized mirror descent** with learning rate $\eta$[2]:

$$\phi^{(t)}(w^{(t+1)}) = \phi^{(t)}(w^{(t)}) - \eta \nabla f_{i_t}(w^{(t)}), \tag{2}$$

where $i_t \in [n]$ is chosen uniformly at random. As described in the introduction, the above algorithm generalizes both gradient descent (where $\phi(x) = x$) and mirror descent (where $\phi^{(t)}(x) = \nabla \psi(x)$ for some strictly convex potential function $\psi$). In the case where $\phi^{(t)}(x) = G^{(t)}x$ for an invertible matrix $G^{(t)} \in \mathbb{R}^{d \times d}$, the update rule in equation (2) reduces to:

$$w^{(t+1)} = w^{(t)} - \eta G^{(t)-1} \nabla f_{i_t}(w^{(t)})$$

Hence, when $\phi^{(t)}$ is an invertible linear transformation, Equation (2) is equivalent to pre-conditioned gradient descent. We now present the Polyak-Lojasiewicz inequality and lemmas from optimization theory that will be used in our proofs[3].

**Polyak-Lojasiewicz (PL) Inequality.** *A function $f : \mathbb{R}^d \to \mathbb{R}$ is $\mu$-PL if for some $\mu > 0$:*

$$\frac{1}{2}\|\nabla f(x)\|^2 \geq \mu(f(x) - f(x^*)) \ \ \forall x \in \mathbb{R}^d, \tag{3}$$

*where $x^* \in \mathbb{R}^d$ is a global minimizer for $f$.*

A useful variation of the PL inequality is the PL[*] inequality introduced in Liu et al. (2020) which does not require knowledge of $f(x^*)$.

**Definition.** *A function $f : \mathbb{R}^d \to \mathbb{R}$ is $\mu$-PL[*] if for some $\mu > 0$:*

$$\frac{1}{2}\|\nabla f(x)\|^2 \geq \mu f(x) \ \ \forall x \in \mathbb{R}^d, \tag{4}$$

A function that is $\mu$-PL[*] is also $\mu$-PL when $f$ is non-negative. Additionally, we will typically assume that $f$ is $L$-smooth (with $L$-Lipschitz continuous derivative).

**Definition.** *A function $f : \mathbb{R}^d \to \mathbb{R}$ is $L$-smooth for $L > 0$ if for all $x, y \in \mathbb{R}^d$:*

$$\|\nabla f(x) - \nabla f(y)\| \leq L\|x - y\|.$$

If $\phi^{(t)}(x) = x$ for any $t$ and $x \in \mathbb{R}^d$ then SGMD reduces to SGD. If $f$ is $L$-smooth and satisfies the PL-Inequality, then SGD converges linearly to a global minimum (Bassily et al., 2018; Karimi et al., 2016; Vaswani et al., 2019). Moreover, the following lemma (proven in Appendix A) shows that the PL[*] condition implies the existence of a global minimum $x^*$ for non-negative, $L$-smooth $f$.

**Lemma 1.** *If $f : \mathbb{R}^d \to \mathbb{R}$ is $\mu$-PL[*], $L$-smooth and $f(x) \geq 0$ for all $x \in \mathbb{R}^d$, then gradient descent with learning rate $\eta < \frac{2}{L}$ converges linearly to $x^*$ satisfying $f(x^*) = 0$.*

---

[2]The framework also allows for adaptive learning rates by using $\eta^{(t)}$ to denote a time-dependent step size.

[3]We assume all norms are the 2-norm unless stated otherwise.

Hence, in cases where the loss function is nonnegative (for example the squared loss), we can remove the usual assumption about the existence of a global minimum, $x^*$, and instead assume that $f$ satisfies the PL$^*$ inequality. We now reference standard properties of $L$-smooth functions (Zhou, 2018), which will be used in our proofs.

**Lemma 2.** *If $f : \mathbb{R}^d \to \mathbb{R}$ is $L$-smooth, then for all $x, y \in \mathbb{R}^d$:*

$$(a)\ \ f(y) \leq f(x) + \langle \nabla f(x), y - x \rangle + \frac{L}{2}\|y - x\|^2,$$

$$(b)\ \ \|\nabla f(x)\|^2 \leq 2L(f(x) - f(x^*)).$$

The following lemma relates $\mu$ and $L$ (the proof is in Appendix B).

**Lemma 3.** *If $f : \mathbb{R}^d \to \mathbb{R}$ is $\mu$-PL and $L$-smooth, then $\mu \leq L$.*

Using Lemma 2b in place of the strong growth condition (i.e. $\mathbb{E}_i[\|\nabla f_i(x)\|^2] \leq \rho \|\nabla f(x)\|^2$) yields slightly different learning rates when establishing convergence of stochastic descent methods (as is apparent from the different learning rates between Bassily et al. (2018) and Vaswani et al. (2019)). The following simple lemma will be used in the proof of Theorem 3.

**Lemma 4.** *If $f(x) = \frac{1}{n}\sum_{i=1}^{n} f_i(x)$ where $f_i : \mathbb{R}^d \to \mathbb{R}$ are $L_i$-smooth , then $f$ is $\sup_i L_i$-smooth.*

Note that there could exist some other constant $L' < \sup_i L_i$ for which $f$ is $L'$-smooth, but this upper bound suffices for our proof of Theorem 3. Lastly, we define and reference standard properties of strongly convex functions (Zhou, 2018), which will be useful in demonstrating how our GMD results generalize those for mirror descent.

**Definition.** *For $\alpha > 0$, a differentiable function, $\psi : \mathbb{R}^d \to \mathbb{R}$, is $\alpha$-strongly convex if for all $x, y$,*

$$\psi(y) \geq \psi(x) + \langle \nabla\psi(x), y - x \rangle + \frac{\alpha}{2}\|y - x\|^2.$$

**Lemma 5.** *If $\psi : \mathbb{R}^d \to \mathbb{R}$ is $\alpha$-strongly convex, then for all $x, y$:*

$$\psi(y) \leq \psi(x) + \langle \nabla\psi(x), y - x \rangle + \frac{1}{2\alpha}\|\nabla\psi(y) - \nabla\psi(x)\|^2.$$

With these preliminaries in hand, we now present our proofs for linear convergence of SGMD using the PL-Inequality.

## 4 SUFFICIENT CONDITIONS FOR LINEAR CONVERGENCE OF SGMD

In this section, we provide sufficient conditions to establish (expected) linear convergence for (stochastic) GMD. We first provide simple conditions under which GMD converges linearly by extending the proof strategy from Karimi et al. (2016). We then present alternate conditions for linear convergence of GMD, which can be naturally extended to the stochastic setting.

### 4.1 SIMPLE CONDITIONS FOR LINEAR CONVERGENCE OF GMD

We begin with a simple set of conditions under which (non-stochastic) GMD converges linearly (the full proof is presented in Appendix C). The main benefit of this analysis is that it is a straightforward extension of the proof of linear convergence for gradient descent under the PL-Inequality presented in Karimi et al. (2016).

**Theorem 1.** *Suppose $f : \mathbb{R}^d \to \mathbb{R}$ is $L$-smooth and $\mu$-PL and $\phi^{(t)} : \mathbb{R}^d \to \mathbb{R}^d$ is an invertible, $\alpha_u^{(t)}$-Lipschitz function where $\lim_{t\to\infty} \alpha_u^{(t)} < \infty$. If for all $x, y \in \mathbb{R}^d$ and for all timesteps $t$ there exist $\alpha_l^{(t)} > 0$ such that*

$$\langle \phi^{(t)}(x) - \phi^{(t)}(y), x - y \rangle \geq \alpha_l^{(t)}\|x - y\|^2,$$

*and $\lim_{t\to\infty} \alpha_l^{(t)} > 0$, then generalized mirror descent converges linearly to a global minimum for any $\eta^{(t)} < \frac{2\alpha_l^{(t)}}{L}$.*

**Remark.** Theorem 1 yields a fixed learning rate provided that $\alpha_l^{(t)}$ is uniformly bounded. In addition, note that Theorem 1 applies also under weaker assumptions, namely when $\phi^{(t)}$ is locally Lipschitz. Finally, the provided learning rate can be computed exactly for settings such as linear regression, since it only requires knowledge of $L$ and $\alpha_l^{(t)}$ (see Section 7). When $\eta = \frac{\alpha_l^{(t)}}{L}$ and given $w^*$ a minimizer of $f$, the proof of Theorem 1 implies that:

$$f(w^{(t+1)}) - f(w^*) \leq \left(1 - \frac{\mu \alpha_l^{(t)2}}{L \alpha_u^{(t)2}}\right)(f(w^{(t)}) - f(w^*)).$$

Letting $\kappa^{(t)} = \frac{L \alpha_u^{(t)2}}{\mu \alpha_l^{(t)2}}$ thus generalizes the condition number introduced in Definition 4.1 of Liu et al. (2020) for gradient descent. Provided that $\kappa = \lim_{t \to \infty} \kappa^{(t)} > 0$, then Theorem 1 guarantees linear convergence to a global minimum. When $\frac{\alpha_l^{(t)}}{\alpha_u^{(t)}}$ is decreasing in $t$, the rate is given by:

$$f(w^{(t+1)}) - f(w^*) \leq \left(1 - \frac{1}{\kappa}\right)^{t+1}(f(w^{(0)}) - f(w^*)).$$

### 4.2 Taylor Series Analysis for Linear Convergence in GMD

Although the proof of Theorem 1 is succinct, it is nontrivial to extend to the stochastic setting[4]. In order to develop a convergence result for the stochastic setting, we turn to an alternate set of conditions for linear convergence by using the Taylor expansion of $\phi^{-1}$. We use $\mathbf{J}_\phi$ to denote the Jacobian of $\phi$. For ease of notation, we consider non-time-dependent $\alpha_l, \alpha_u$, but our results are trivially extendable to the setting when these quantities are time-dependent.

**Theorem 2.** *Suppose $f : \mathbb{R}^d \to \mathbb{R}$ is $L$-smooth and $\mu$-PL and $\phi : \mathbb{R}^d \to \mathbb{R}^d$ is an infinitely differentiable, analytic function with analytic inverse, $\phi^{-1}$. If there exist $\alpha_l, \alpha_u > 0$ such that*

$(a)\ \alpha_l \mathbf{I} \preccurlyeq \mathbf{J}_\phi \preccurlyeq \alpha_u \mathbf{I},$

$(b)\ |\partial_{i_1, \ldots i_k} \phi_j^{-1}(x)| \leq \dfrac{k!}{2 \alpha_u d}\ \ \forall x \in \mathbb{R}^d, i_1, \ldots i_k \in [d], j \in [d], k \geq 2,$

*then generalized mirror descent converges linearly for any $\eta^{(t)} < \min\left(\frac{4\alpha_l^2}{5L\alpha_u}, \frac{1}{2\sqrt{d}\|\nabla f(w^{(t)})\|}\right)$.*

The full proof is provided in Appendix D. Importantly, the adaptive component of the learning rate is only used to ensure that the sum of the higher order terms for the Taylor expansion converges. In particular, if $\phi^{(t)}$ is a linear function, then our learning rate no longer needs to be adaptive. Note that alternatively, we can establish linear convergence for a fixed learning rate given that the gradients monotonically decrease or if $f$ is non-negative and $\mu$-PL$^*$. We analyze this case in Appendix E and provide an explicit condition on $\mu$ and $L$ under which this holds.

### 4.3 Taylor Series Analysis for Linear Convergence in Stochastic GMD

The main benefit of the above Taylor series analysis is that it naturally extends to the stochastic setting as demonstrated in the following result (with proof presented in Appendix F).

**Theorem 3.** *Suppose $f(x) = \frac{1}{n} \sum_{i=1}^n f_i(x)$ where $f_i : \mathbb{R}^d \to \mathbb{R}$ are non-negative, $L_i$-smooth functions with $L = \sup_{i \in [n]} L_i$ and $f$ is $\mu$-PL$^*$. Let $\phi : \mathbb{R}^d \to \mathbb{R}^d$ be an infinitely differentiable, analytic function with analytic inverse, $\phi^{-1}$. SGMD is used to minimize $f$ according to the updates:*

$$\phi(w^{(t+1)}) = \phi(w^{(t)}) - \eta^{(t)} \nabla f_{i_t}(w^{(t)}),$$

*where $i_t \in [n]$ is chosen uniformly at random and $\eta^{(t)}$ is an adaptive step size. If there exist $\alpha_l, \alpha_u > 0$ such that:*

$(a)\ \alpha_l \mathbf{I} \preccurlyeq \mathbf{J}_\phi \preccurlyeq \alpha_u \mathbf{I},$

$(b)\ |\partial_{i_1, \ldots i_k} \phi_j^{-1}(x)| \leq \dfrac{k!\,\mu}{2\alpha_u d L}\ \ \forall x \in \mathbb{R}^d, i_1, \ldots i_k \in [d], j \in [d], k \geq 2,$

*then SGMD with $\eta^{(t)} < \min\left(\frac{4\mu\alpha_l^2}{5L^2\alpha_u}, \frac{1}{2\sqrt{d}\max_i \|\nabla f_i(w^{(t)})\|}\right)$ converges linearly to a global minimum.*

---

[4]The main difficulty is relating $w^{(t+1)} - w^{(t)}$ to the gradient at timestep $t$.

**Remark.** Note that there is a slight difference between the learning rate in Theorem 2 and Theorem 3 due to a multiplicative factor of $\mu$. Consistent with the difference in learning rates between Bassily et al. (2018) and Vaswani et al. (2019), we can make the learning rate between the two theorems match if we assume the strong growth condition (i.e. $\mathbb{E}_i[\|\nabla f_i(x)\|^2] \leq \rho \|\nabla f(x)\|^2$) with $\rho = \mu$ instead of using Lemma 2b. Moreover, as $\max_i \|\nabla f_i(w^{(t)})\| \leq \sqrt{2nLf(w^{(0)})}$, we establish linear convergence for a fixed step size $\eta < \min\left( \frac{4\mu\alpha_l^2}{5L^2\alpha_u}, \frac{1}{2\sqrt{2dnLf(w^{(0)})}} \right)$ as well.

## 5 COROLLARIES OF LINEAR CONVERGENCE IN SGMD

We now present how the linear convergence results established by Theorems 1, 2, and 3 apply to commonly used optimization algorithms including mirror descent and Adagrad. In this section, we primarily extend the analysis from Theorem 1 for the non-stochastic case. However, our results can be extended analogously to give expected linear convergence in the stochastic case by using the extension provided in Theorem 3.

**Gradient Descent.** For the case of gradient descent, $\phi(x) = x$ and so $\alpha_l = \alpha_u = 1$. Hence, we see that gradient descent converges linearly under the conditions of Theorem 1 with $\eta < \frac{2}{L}$, which is consistent with the analysis in Karimi et al. (2016).

**Mirror Descent.** Let $\psi : \mathbb{R}^d \to \mathbb{R}$ be a strictly convex potential. Thus, $\phi(x) = \nabla\psi(x)$ is an invertible function. If $\psi$ is $\alpha_l$-strongly convex and (locally) $\alpha_u$-Lipschitz and $f$ is $L$-smooth and $\mu$-PL, then the conditions of Theorem 1 are satisfied. Moreover, since the $\alpha_u$-Lipschitz condition holds locally for most potentials considered in practice, our result implies linear convergence for mirror descent with $\alpha_l$-strongly convex potential $\psi$.

**Adagrad.** Let $\phi^{(t)} = \mathcal{G}^{(t)\frac{1}{2}}$ where $\mathcal{G}^{(t)}$ is a diagonal matrix such that

$$\mathcal{G}_{i,i}^{(t)} = \sum_{k=0}^{t} \nabla f_i(w^{(k)})^2.$$

Then GMD corresponds to Adagrad. In this case, we can apply Theorem 1 to establish linear convergence of Adagrad under the PL-Inequality provided that $\phi^{(t)}$ satisfies the condition of Theorem 1. The following corollary proves that this condition holds and hence that Adagrad converges linearly.

**Corollary 1.** *Let $f : \mathbb{R}^d \to \mathbb{R}$ be an $L$-smooth function that is $\mu$-PL. Let $\alpha_l^{(t)^2} = \min_{i \in [d]} \mathcal{G}_{i,i}^{(t)}$ and $\alpha_u^{(t)^2} = \max_{i \in [d]} \mathcal{G}_{i,i}^{(t)}$. If $\lim_{t\to\infty} \frac{\alpha_l^{(t)}}{\alpha_u^{(t)}} \neq 0$, then Adagrad converges linearly for adaptive step size $\eta^{(t)} = \frac{\alpha_l^{(t)}}{L}$.*

The proof is presented in Appendix H. While Corollary 1 can be extended to the stochastic setting via Theorem 3, it requires knowledge of $\mu$ to setup the learning rate, and the resulting learning rate provided is typically smaller than what we can use in practice. We analyze this case further in Section 7. Additionally, since the condition $\lim_{t\to\infty} \frac{\alpha_l^{(t)}}{\alpha_u^{(t)}} \neq 0$ is difficult to verify in practice, we provide Corollary 2 in Appendix H, which presents a verifiable condition under which Adagrad converges linearly.

## 6 LOCAL CONVERGENCE AND IMPLICIT REGULARIZATION IN GMD

In the previous sections, we established linear convergence for GMD for real-valued loss, $f : \mathbb{R}^d \to \mathbb{R}$, that is $\mu$-PL for all $x \in \mathbb{R}^d$. In this section, we show that $f$ need only satisfy the PL inequality locally (i.e. within a ball of fixed radius around the initialization) in order to establish linear convergence. The following theorem (proof in Appendix G) extends Theorem 4.2 from Liu et al. (2020) to GMD and uses the PL* condition to establish both the existence of a global minimum and linear convergence to this global minimum under GMD[5]. We use $\mathcal{B}(x, R) = \{z ; z \in \mathbb{R}^d, \|x - z\|_2 \leq R\}$ to denote the ball of radius $R$ centered at $x$.

---

[5]We require additional assumptions on $\phi^{(t)}$ for the case of time-dependent mirrors (see Appendix G.)

**Theorem 4.** *Suppose* $\phi : \mathbb{R}^d \to \mathbb{R}^d$ *is an invertible,* $\alpha_u$-*Lipschitz function and that* $f : \mathbb{R}^d \to \mathbb{R}$ *is non-negative,* $L$-*smooth, and* $\mu$-$PL^*$ *on* $\tilde{\mathcal{B}} = \{x \ ; \ \phi(x) \in \mathcal{B}(\phi(w^{(0)}), R)\}$ *with* $R = \frac{2\sqrt{2L}\sqrt{f(w^{(0)})}\alpha_u^2}{\alpha_l \mu}$. *If for all* $x, y \in \mathbb{R}^d$ *there exists* $\alpha_l > 0$ *such that*

$$\langle \phi(x) - \phi(y), x - y \rangle \geq \alpha_l \|x - y\|^2,$$

*then,*

   (1) *There exists a global minimum* $w^{(\infty)} \in \tilde{\mathcal{B}}$.

   (2) *GMD converges linearly to* $w^{(\infty)}$ *for* $\eta = \dfrac{\alpha_l}{L}$.

   (3) *If* $w^* = \underset{w \in \tilde{\mathcal{B}} \ ; \ f(w)=0}{\arg\min} \|\phi(w) - \phi(w^{(0)})\|$, *then,* $\|\phi(w^*) - \phi(w^{(\infty)})\| \leq 2R$.

**Approximate Implicit Regularization in GMD.** When $R$ is small, we can view the result of Theorem 4 as a characterization of the solution selected by GMD, thereby obtaining approximate implicit regularization results for GMD. Namely, for $\delta = \frac{R}{2}$, we have $\|\phi(w^*) - \phi(w^\infty)\| \leq \delta$. Hence provided that $R$ is small (which holds for small $f(w^{(0)})$), GMD selects an interpolating solution that is close to $w^*$ in $\ell_2$-norm in the dual space induced by $\phi$. This view is consistent with the characterization of approximate implicit regularization in Azizan et al. (2019), as is shown by Corollary 3 in Appendix I. In particular, Corollary 3 implies the assumptions used in Azizan et al. (2019) for the full batch case by proving (1) the existence of such a $w^{(\infty)}$, (2) *linear* convergence of $w^{(0)}$ to $w^{(\infty)}$, and (3) providing explicit forms for $\epsilon$ (where $\epsilon = R^2$ above). Importantly, the approximate implicit regularization result for mirror descent does not need to be stated in terms of Bregman divergence, but can be viewed more naturally as $\|\nabla\psi(w^{(\infty)}) - \nabla\psi(w^*)\|_2$ being small.

## 7    EXPERIMENTAL VERIFICATION OF OUR THEORETICAL RESULTS

We now present a simple set of experiments under which we can explicitly compute the learning rates in our theorems. We will show that in accordance with our theory, both fixed and adaptive versions of these learning rates yield linear convergence. We focus on computing learning rates for Adagrad in the noiseless regression setting used in Xie et al. (2020). Namely, we are given $(X, y) \in \mathbb{R}^{n \times d} \times \mathbb{R}^n$ such that there exists a $w^* \in \mathbb{R}^d$ such that $Xw^* = y$. If $n < d$, then the system is over-parameterized, and if $n \geq d$, the system is sufficiently parameterized and has a unique solution.

In this setting, the squared loss (MSE) is $L$-smooth with $L = \lambda_{\max}(XX^T)$, and it is $\mu$-PL with $\mu = \lambda_{\min}(XX^T)$ where $\lambda_{max}$ and $\lambda_{min}$ refer to the largest and smallest non-zero eigenvalues, respectively[6]. Moreover, for Adagrad, we can compute $\alpha_l^{(t)} = \min_{i \in [d]} (\sum_{k=0}^t \nabla f_i(w^{(k)}))^2$ and $\alpha_u^{(t)} = \max_{i \in [d]} (\sum_{k=0}^t \nabla f_i(w^{(k)}))^2$ at each timestep. Hence for Adagrad in the noiseless linear regression setting, we can explicitly compute the learning rate provided in Theorem 3 for the stochastic setting and in Corollary 1 for the full batch setting.

Figure 1 demonstrates that in both, the over-parameterized and sufficiently parameterized settings, our provided learning rates yield linear convergence. In the stochastic setting, the theory for fixed learning rates suggests a very small rate ($\approx 10^{-9}$ for Figure 1d) and hence we chose to only present the more reasonable adaptive step size as a comparison. In the full batch setting, the learning rate obtained from our theorems out-performs using the standard fixed learning rate of $0.1$, while performance is comparable for the stochastic setting. Interestingly, our theory suggests an adaptive learning rate that is increasing (in contrast to the usual decreasing learning rate schedules). In particular, while the suggested learning rate for Figure 1a starts at $0.99$, it increases to $1.56$ at the end of training.

In Appendix J, we present experiments on over-parameterized neural networks. While the PL-condition holds in this setting (Liu et al., 2020), it can be difficult to compute the smoothness parameter $L$ (which was the motivation for developing Adagrad-Norm). Interestingly, our experiments

---

[6]We take $\mu$ as the smallest non-zero eigenvalue since Adagrad updates keep parameters in the span of the data.

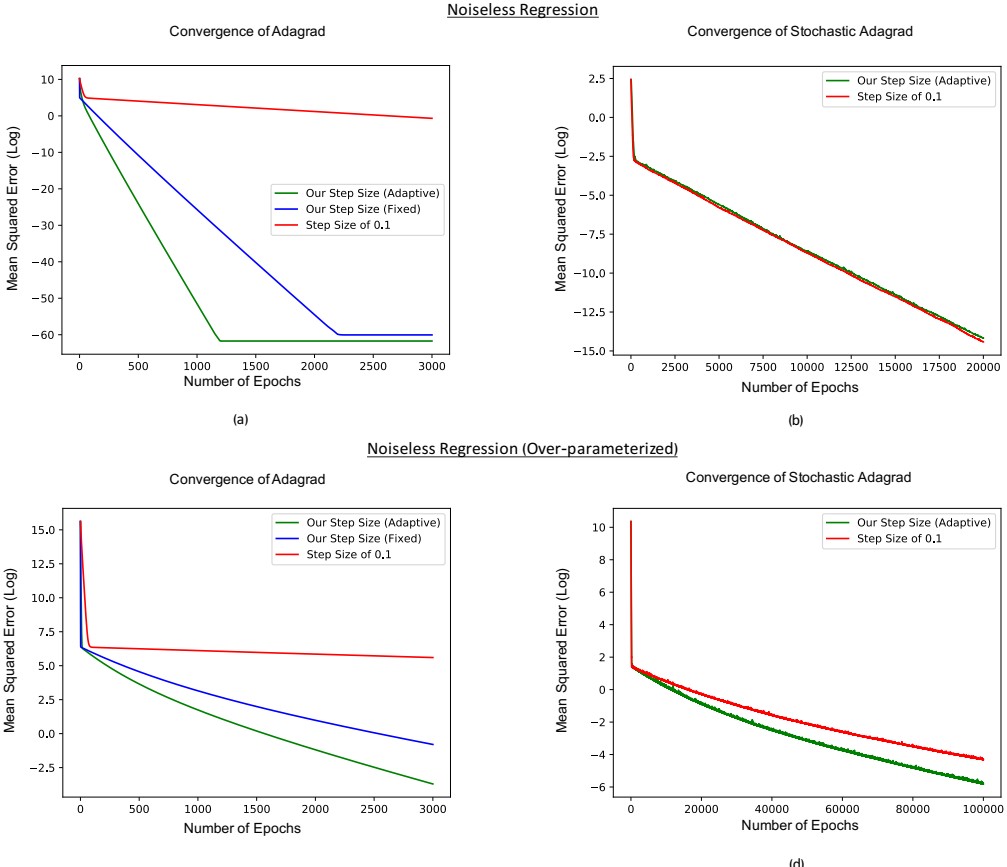

Figure 1: Using the rates provided by Corollary 1 leads to linear convergence for (Stochastic) Adagrad in the noiseless linear regression setting also considered in Xie et al. (2020). (a, b) Noiseless linear regression on 2000 examples in 20 dimensions. (c, d) Noiseless linear regression on 200 examples in 1000 dimensions.

demonstrate that our increasing adaptive learning rate from Theorem 1, using an approximation for $L$, provides convergence for Adagrad in over-parameterized networks. The link to the code is provided in Appendix J.

## 8 CONCLUSION

In this work, we presented stochastic generalized mirror descent, which generalizes both mirror descent and pre-conditioner methods. By using the PL-condition and a Taylor-series based analysis, we provided sufficient conditions for linear convergence of SGMD in the non-convex setting. As a corollary, we obtained sufficient conditions for linear convergence of both mirror descent and pre-conditioner methods such as Adagrad. Lastly, we prove the existence of an interpolating solution and linear convergence of GMD to this solution for non-negative loss functions that are locally $PL^*$. Importantly, our local convergence results allow us to obtain approximate implicit regularization results for GMD. Namely, we prove that GMD linearly converges to an interpolating solution that is approximately the closest interpolating solution to the initialization in $\ell_2$ norm in the dual space. For the full batch setting, this result provides a more natural characterization of implicit regularization in terms of $\ell_2$ norm in the dual space, as opposed to Bregman divergence.

Looking ahead, we envision that the generality of our analysis (and the PL-condition) could provide useful in the analysis of other commonly used adaptive methods such as Adam (Kingma & Ba, 2015). Moreover, since the PL-condition holds in varied settings including over-parameterized neural networks (Liu et al., 2020), it would be interesting to analyze whether the learning rates obtained here provide an improvement for convergence in these modern settings.

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

APPENDIX

## A   PROOF OF LEMMA 1

We restate the lemma below.

**Lemma.** *If $f : \mathbb{R}^d \to \mathbb{R}$ is $\mu$-PL$^*$, L-smooth and $f(x) \geq 0$ for all $x \in \mathbb{R}^d$, then gradient descent with learning rate $\eta < \frac{2}{L}$ converges linearly to $x^*$ satisfying $f(x^*) = 0$.*

*Proof.* The proof follows exactly from Theorem 1 of Karimi et al. (2016). Since $f$ is $L$-smooth, by Lemma 2a it holds that:

$$f(w^{(t+1)}) - f(w^{(t)}) \leq \langle \nabla f(w^{(t)}), w^{(t+1)} - w^{(t)} \rangle + \frac{L}{2} \| w^{(t+1)} - w^{(t)} \|^2.$$

$$\implies f(w^{(t+1)} - f(w^{(t)}) \leq -\eta \| \nabla f(w^{(t)}) \|^2 + \frac{L}{2} \eta^2 \| \nabla f(w^{(t)}) \|^2$$

$$\implies f(w^{(t+1)} - f(w^{(t)}) \leq \left( -\eta + \frac{\eta^2 L}{2} \right) 2\mu f(w^{(t)})$$

$$\implies f(w^{(t+1)}) \leq \left( 1 - 2\mu\eta + \mu\eta^2 L \right) f(w^{(t)})$$

Hence, if $\eta < \frac{2}{L}$, then $C = \left( 1 - 2\mu\eta + \mu\eta^2 L \right) < 1$. Thus, we have $f(w^{(t+1)}) \leq Cf(w^{(t)})$ for $C < 1$. Thus, as $f$ is bounded below by 0 and the sequence $\{f(w^{(t)})\}_{t \in \mathbb{N}}$ monotonically decreases with infimum 0, the monotone convergence theorem implies $\lim_{t \to \infty} f(w^{(t)}) = 0$. □

## B   PROOF OF LEMMA 3

*Proof.* From Lemma 2 and from the PL condition, we have:

$$2\mu(f(x) - f(x^*)) \leq \| \nabla f(x) \|^2 \leq 2L(f(x) - f(x^*)) \implies \mu \leq L \quad □$$

## C   PROOF OF THEOREM 1

*Proof.* Since $f$ is $L$-smooth, by Lemma 2a it holds that:

$$f(w^{(t+1)}) - f(w^{(t)}) \leq \langle \nabla f(w^{(t)}), w^{(t+1)} - w^{(t)} \rangle + \frac{L}{2} \| w^{(t+1)} - w^{(t)} \|^2. \tag{5}$$

Now by the condition on $\phi^{(t)}$ in Theorem 1, we bound the first term on the right as follows:

$$\langle \phi^{(t)}(w^{(t+1)}) - \phi^{(t)}(w^{(t)}), w^{(t+1)} - w^{(t)} \rangle \geq \alpha_l^{(t)} \| w^{(t+1)} - w^{(t)} \|^2$$

$$\implies \langle -\eta \nabla f(w^{(t)}), w^{(t+1)} - w^{(t)} \rangle \geq \alpha_l^{(t)} \| w^{(t+1)} - w^{(t)} \|^2 \text{ using Equation (2)}$$

$$\implies \langle \nabla f(w^{(t)}), w^{(t+1)} - w^{(t)} \rangle \leq -\frac{\alpha_l^{(t)}}{\eta} \| w^{(t+1)} - w^{(t)} \|^2.$$

Substituting this bound back into the inequality in (5), we obtain

$$f(w^{(t+1)}) - f(w^{(t)}) \leq \left( -\frac{\alpha_l^{(t)}}{\eta} + \frac{L}{2} \right) \| w^{(t+1)} - w^{(t)} \|^2.$$

Since the learning rate is selected so that the coefficient of $\|w^{(t+1)} - w^{(t)}\|^2$ on the right is negative, we obtain

$$f(w^{(t+1)}) - f(w^{(t)}) \leq \left(-\frac{\alpha_l^{(t)}}{\eta} + \frac{L}{2}\right) \|w^{(t+1)} - w^{(t)}\|^2$$

$$\leq \left(-\frac{\alpha_l^{(t)}}{\eta} + \frac{L}{2}\right) \frac{1}{\alpha_u^{(t)2}} \|\phi^{(t)}(w^{(t+1)}) - \phi^{(t)}(w^{(t)})\|^2$$

$$= \left(-\frac{\alpha_l^{(t)}}{\eta} + \frac{L}{2}\right) \frac{1}{\alpha_u^{(t)2}} \|-\eta\nabla f(w^{(t)})\|^2 \quad \text{using Equation (1)}$$

$$\leq \left(-\frac{\alpha_l^{(t)}}{\eta} + \frac{L}{2}\right) 2\mu \frac{\eta^2}{\alpha_u^{(t)2}} (f(w^{(t)}) - f(w^*)) \quad \text{as } f \text{ is } \mu\text{-PL}$$

$$\implies f(w^{(t+1)}) - f(w^*) \leq \left(1 - 2\mu\frac{\eta\alpha_l^{(t)}}{\alpha_u^{(t)2}} + \mu\frac{L\eta^2}{\alpha_u^{(t)2}}\right) (f(w^{(t)}) - f(w^*)),$$

where the second inequality follows since $\phi^{(t)}$ is $\alpha_u^{(t)}$-Lipschitz. For linear convergence, we need.

$$0 < 1 - 2\mu\frac{\eta\alpha_l^{(t)}}{\alpha_u^{(t)2}} + \mu\frac{L\eta^2}{\alpha_u^{(t)2}} < 1. \tag{6}$$

From Lemma 3, $\mu < \frac{\alpha_u^{(t)2}L}{\alpha_l^{(t)}}$ always holds and implies that the left inequality in (6) is satisfied for all $\eta^{(t)}$. The right inequality holds by our assumption that $\eta^{(t)} < \frac{2\alpha_l^{(t)}}{L}$, which completes the proof. $\quad\square$

## D  PROOF OF THEOREM 2

We repeat the theorem below for convenience.

**Theorem.** *Suppose $f : \mathbb{R}^d \to \mathbb{R}$ is $L$-smooth and $\mu$-PL and $\phi : \mathbb{R}^d \to \mathbb{R}^d$ is an infinitely differentiable, analytic function with analytic inverse, $\phi^{-1}$. If there exist $\alpha_l, \alpha_u > 0$ such that:*

*(a)  $\alpha_l \mathbf{I} \preccurlyeq \mathbf{J}_\phi \preccurlyeq \alpha_u \mathbf{I}$,*

*(b)  $|\partial_{i_1,\ldots i_k}\phi_j^{-1}(x)| \leq \dfrac{k!}{2\alpha_u d} \quad \forall x \in \mathbb{R}^d, i_1, \ldots i_k \in [d], j \in [d], k \geq 2,$*

*then generalized mirror descent converges linearly for $\eta^{(t)} < \min\left(\frac{4\alpha_l^2}{5L\alpha_u}, \frac{1}{2\sqrt{d}\|\nabla f(w^{(t)})\|}\right)$.*

*Proof.* Since $f$ is $L$-smooth, it holds by Lemma that 2:

$$f(w^{(t+1)}) - f(w^{(t)}) \leq \langle \nabla f(w^{(t)}), w^{(t+1)} - w^{(t)}\rangle + \frac{L}{2}\|w^{(t+1)} - w^{(t)}\|^2.$$

Next, we want to bound the two quantities on the right hand side by a multiple of $\|\nabla f(w^{(t)})\|^2$. We do so by expanding $w^{(t+1)} - w^{(t)}$ using the Taylor series for $\phi^{-1}$ as follows:

$$w^{(t+1)} - w^{(t)} = \phi^{-1}(\phi(w^{(t)}) - \eta\nabla f(w^{(t)})) - w^{(t)}$$

$$= -\eta\mathbf{J}_{\phi^{-1}}(\phi(w^{(t)}))\nabla f(w^{(t)})$$

$$+ \sum_{k=2}^{\infty} \frac{1}{k!}\left[\sum_{i_1,i_2\ldots i_k=1}^{d}(-\eta)^k\partial_{i_1,\ldots i_k}\phi_j^{-1}(\phi(w^{(t)}))(\nabla f(w^{(t)})_{i_1}\ldots\nabla f(w^{(t)})_{i_k})\right].$$

The quantity in brackets is a column vector where we only wrote out the $j^{th}$ coordinate for $j \in [d]$. Now we bound the term $\langle \nabla f(w^{(t)}), w^{(t+1)} - w^{(t)}\rangle$:

$$\langle \nabla f(w^{(t)}), w^{(t+1)} - w^{(t)}\rangle = -\eta\nabla f(w^{(t)})^T\mathbf{J}_\phi^{-1}(w^{(t)})\nabla f(w^{(t)})$$

$$+ \nabla f(w^{(t)})^T\sum_{k=2}^{\infty}\frac{1}{k!}\left[\sum_{i_1,i_2\ldots i_k=1}^{d}(-\eta)^k\partial_{i_1,\ldots i_k}\phi_j^{-1}(\phi(w^{(t)}))(\nabla f(w^{(t)})_{i_1}\ldots\nabla f(w^{(t)})_{i_k})\right].$$

We have separated the first order term from the other orders because we will bound them separately using conditions (a) and (b) respectively. Namely, we first have:

$$-\eta \nabla f(w^{(t)})^T \mathbf{J}_\phi^{-1}(w^{(t)}) \nabla f(w^{(t)}) \le -\frac{\eta}{\alpha_u} \|\nabla f(w^{(t)})\|^2.$$

Next, we use the Cauchy-Schwarz inequality on inner products to bound the inner product of $\nabla f(w^{(t)})$ and the higher order terms. In the following, we use $\alpha$ to denote $\frac{1}{2\alpha_u d}$.

$$\nabla f(w^{(t)})^T \sum_{k=2}^{\infty} \frac{1}{k!} \left[ \sum_{i_1,i_2\dots i_k=1}^d (-\eta)^k \partial_{i_1,\dots i_k} \phi_j^{-1}(\phi(w^{(t)}))(\nabla f(w^{(t)})_{i_1} \dots \nabla f(w^{(t)})_{i_k}) \right]$$

$$\le \|\nabla f(w^{(t)})\| \sum_{k=2}^{\infty} \frac{1}{k!} \left\| \left[ \sum_{i_1,i_2\dots i_k=1}^d (-\eta)^k \partial_{i_1,\dots i_k} \phi_j^{-1}(\phi(w^{(t)}))(\nabla f(w^{(t)})_{i_1} \dots \nabla f(w^{(t)})_{i_k}) \right] \right\|$$

$$\le \|\nabla f(w^{(t)})\| \sum_{k=2}^{\infty} \frac{\alpha k!}{k!} (\eta)^k \left\| \left[ \sum_{i_1,i_2\dots i_k=1}^d (|\nabla f(w^{(t)})_{i_1}| \dots |\nabla f(w^{(t)})_{i_k}|) \right] \right\|$$

$$= \|\nabla f(w^{(t)})\| \alpha \sum_{k=2}^{\infty} \sqrt{d}(\eta)^k (|\nabla f(w^{(t)})_1| + \dots |\nabla f(w^{(t)}))_d|)^k$$

$$= \|\nabla f(w^{(t)})\| \alpha \sum_{k=2}^{\infty} (\eta)^k \sqrt{d} |\langle \begin{bmatrix} |\nabla f(w^{(t)})_1| \\ \vdots \\ |\nabla f(w^{(t)})_d| \end{bmatrix}, \mathbf{1} \rangle|^k$$

$$\le \|\nabla f(w^{(t)})\| \alpha \sum_{k=2}^{\infty} (\eta)^k \sqrt{d} \|\nabla f(w^{(t)})\|^k (\sqrt{d})^k$$

$$= \alpha \sum_{k=2}^{\infty} (\sqrt{d})^{k+1} (\eta)^k \|\nabla f(w^{(t)})\|^{k+1}$$

$$= \alpha(\sqrt{d})^3(\eta)^2 \|\nabla f(w^{(t)})\|^3 \sum_{k=0}^{\infty} (\sqrt{d})^k (\eta)^k \|\nabla f(w^{(t)})\|^k = \frac{\alpha(\sqrt{d})^3(\eta)^2 \|\nabla f(w^{(t)})\|^3}{1 - \sqrt{d}\eta \|\nabla f(w^{(t)})\|}.$$

Hence we can select $\eta < \frac{1}{2\sqrt{d}\|\nabla f(w^{(t)})\|}$ such that:

$$\frac{\alpha(\sqrt{d})^3(\eta)^2 \|\nabla f(w^{(t)})\|^3}{1 - \sqrt{d}\eta \|\nabla f(w^{(t)})\|} \le \frac{\alpha(\sqrt{d})^3(\eta)^2 \|\nabla f(w^{(t)})\|^3}{\sqrt{d}\eta \|\nabla f(w^{(t)})\|} = d\alpha\eta \|\nabla f(w^{(t)})\|^2.$$

Thus, we have established the following bound:

$$\langle \nabla f(w^{(t)}), w^{(t+1)} - w^{(t)} \rangle \le \left( -\frac{\eta}{\alpha_u} + d\alpha\eta \right) \|\nabla f(w^{(t)})\|^2 = \left( -\frac{\eta}{2\alpha_u} \right) \|\nabla f(w^{(t)})\|^2.$$

Proceeding analogously as above, we establish a bound on $\|w^{(t+1)} - w^{(t)}\|^2$:

$$\|w^{(t+1)} - w^{(t)}\|^2 \le \left( \frac{\eta^2}{\alpha_l^2} + \alpha^2 d^2 \eta^2 \right) \|\nabla f(w^{(t)})\|^2 = \left( \frac{\eta^2}{\alpha_l^2} + \frac{\eta^2}{4\alpha_u^2} \right) \|\nabla f(w^{(t)})\|^2.$$

Putting the bounds together we obtain:

$$f(w^{(t+1)}) - f(w^{(t)}) \le \left( -\frac{\eta}{2\alpha_u} + \frac{L\eta^2}{2\alpha_l^2} + \frac{L\eta^2}{8\alpha_u^2} \right) \|\nabla f(w^{(t)})\|^2.$$

We select our learning rate to make the coefficient of $\|\nabla f(w^{(t)}\|^2$ negative, and thus by the PL-inequality (4), we have:

$$f(w^{(t+1)}) - f(w^{(t)}) \le \left( -\frac{\eta}{2\alpha_u} + \frac{L\eta^2}{2\alpha_l^2} + \frac{L\eta^2}{8\alpha_u^2} \right) 2\mu(f(w^{(t)}) - f(w^*))$$

$$\implies f(w^{(t+1)}) - f(w^*) \le \left( 1 - \frac{\mu\eta}{\alpha_u} + \frac{\mu L\eta^2}{\alpha_l^2} + \frac{\mu L\eta^2}{4\alpha_u^2} \right) (f(w^{(t)}) - f(w^*)).$$

Hence, $w^{(t)}$ converges linearly when:

$$0 < 1 - \frac{\mu\eta}{\alpha_u} + \frac{\mu L\eta^2}{\alpha_l^2} + \frac{\mu L\eta^2}{4\alpha_u^2} < 1.$$

To show that the left hand side is true, we analyze when the discriminant is negative. Namely, we have that the left side holds if:

$$\frac{\mu^2}{\alpha_u^2} - \frac{4\mu L}{\alpha_l^2} - \frac{\mu L}{\alpha_u^2} < 0$$

$$\implies \frac{\mu}{\alpha_u^2} < \frac{4L}{\alpha_l^2} + \frac{L}{\alpha_u^2}$$

$$\implies \mu < \frac{4L\alpha_u^2}{\alpha_l^2} + L.$$

Since $\mu < L$ by Lemma 3, this is always true. The right hand side holds when $\eta < \frac{4\alpha_l^2}{5L\alpha_u}$, which holds by the assumption of the theorem, thereby completing the proof. □

Note that if $f$ is non-negative and $\mu$-PL$^*$, then we have:

$$\eta^{(t)} \leq \frac{1}{2\sqrt{2Ld}\sqrt{f(w^{(0)})}} \leq \frac{1}{2\sqrt{2Ld}\sqrt{f(w^{(t)})}} \leq \frac{1}{2\sqrt{d}\|\nabla f(w^{(t)})\|}$$

Hence, we can use a fixed learning rate of $\eta = \min\left(\frac{4\alpha_l^2}{5L\alpha_u}, \frac{1}{2\sqrt{2Ld}\sqrt{f(w^{(0)})}}\right)$ in this setting.

## E  CONDITIONS FOR MONOTONICALLY DECREASING GRADIENTS

As discussed in the remarks after Theorem 2, we can provide a fixed learning rate for linear convergence provided that the gradients are monotonically decreasing. As we show below, this requires special conditions on the PL constant, $\mu$, and the smoothness constant, $L$, for $f$.

**Proposition 1.** *Suppose $f : \mathbb{R}^d \to \mathbb{R}$ is $L$-smooth and $\mu$-PL and $\phi : \mathbb{R}^d \to \mathbb{R}^d$ is an infinitely differentiable, analytic function with analytic inverse, $\phi^{-1}$. If there exist $\alpha_l, \alpha_u > 0$ such that:*

*(a)* $\alpha_l \mathbf{I} \preccurlyeq \mathbf{J}_\phi \preccurlyeq \alpha_u \mathbf{I}$,

*(b)* $|\partial_{i_1,\dots i_k} \phi_j^{-1}(x)| \leq \dfrac{k!}{2\alpha_u d} \quad \forall x \in \mathbb{R}^d, i_1, \dots i_k \in [d], j \in [d], k \geq 2$,

*(c)* $\dfrac{\mu}{L} > \dfrac{4\alpha_u^2 + \alpha_l^2}{4\alpha_u^2 + 2\alpha_l^2}$,

*then generalized mirror descent converges linearly for any $\eta < \min\left(\frac{4\alpha_l^2}{5L\alpha_u}, \frac{1}{2\sqrt{d}\|\nabla f(w^{(0)})\|}\right)$.*

*Proof.* Let $C = 1 - \frac{\mu\eta}{\alpha_u} + \frac{\mu L\eta^2}{\alpha_l^2} + \frac{\mu L\eta^2}{4\alpha_u^2}$. We follow exactly the proof of Theorem 2 except that at each timestep we need $C < \frac{\mu}{L}$ (which is less than 1 by Lemma 3) in order for the gradients to converge monotonically since:

$$\|\nabla f(w^{(t+1)})\|^2 \leq 2L(f(w^{(t+1)}) - f(w^*)) \quad \text{See Lemma 2}$$

$$\leq 2LC(f(w^{(t)}) - f(w^*))$$

$$\leq \frac{LC}{\mu}\|\nabla f(w^{(t)})\|^2 \quad \text{As } f \text{ is } \mu\text{-PL}.$$

Hence in order for $\|\nabla f(w^{(t+1)})\|^2 < \|\nabla f(w^{(t)})\|^2$, we need $C < \frac{\mu}{L}$. Thus, we select our learning rate such that:

$$0 < 1 - \frac{\mu\eta}{\alpha_u} + \frac{\mu L\eta^2}{\alpha_l^2} + \frac{\mu L\eta^2}{4\alpha_u^2} < \frac{\mu}{L}.$$

Now, in order to have a solution to this system, we must ensure that the discriminant of the quadratic equation in $\eta$ when considering the right hand side inequality is larger than zero. In particular we require:

$$\frac{\mu^2}{\alpha_u^2} - 4\left(1 - \frac{\mu}{L}\right)\left(\frac{\mu L}{\alpha_l^2} + \frac{\mu L}{4\alpha_u^2}\right) > 0$$

$$\implies \frac{\mu}{L} > \frac{4\alpha_u^2 + \alpha_l^2}{4\alpha_u^2 + 2\alpha_l^2},$$

which completes the proof. $\qquad\square$

## F  Proof of Theorem 3

We repeat the theorem below for convenience.

**Theorem.** *Suppose $f(x) = \frac{1}{n}\sum_{i=1}^{n} f_i(x)$ where $f_i : \mathbb{R}^d \to \mathbb{R}$ are non-negative, $L_i$-smooth functions with $L = \sup_{i \in [n]} L_i$ and $f$ is $\mu$-PL$^*$. Let $\phi : \mathbb{R}^d \to \mathbb{R}^d$ be an infinitely differentiable, analytic function with analytic inverse, $\phi^{-1}$. SGMD is used to minimize $f$ according to the updates:*

$$\phi(w^{(t+1)}) = \phi(w^{(t)}) - \eta^{(t)}\nabla f_{i_t}(w^{(t)}),$$

*where $i_t \in [n]$ is chosen uniformly at random and $\eta^{(t)}$ is an adaptive step size. If there exist $\alpha_l, \alpha_u > 0$ such that:*

*(a) $\alpha_l \mathbf{I} \preccurlyeq \mathbf{J}_\phi \preccurlyeq \alpha_u \mathbf{I}$,*

*(b) $|\partial_{i_1,\ldots i_k}\phi_j^{-1}(x)| \leq \dfrac{k!\,\mu}{2\alpha_u dL}\ \ \forall x \in \mathbb{R}^d, i_1,\ldots i_k \in [d], j \in [d], k \geq 2,$*

*then SGMD converges linearly to a global minimum for any $\eta^{(t)} < \min\left(\dfrac{4\mu\alpha_l^2}{5L^2\alpha_u}, \dfrac{1}{2\sqrt{d}\max_i\|\nabla f_i(w^{(t)})\|}\right)$.*

*Proof.* We follow the proof of Theorem 2. Namely, Lemma 4 implies that $f$ is $L$-smooth and hence

$$f(w^{(t+1)}) - f(w^{(t)}) \leq \langle\nabla f(w^{(t)}), w^{(t+1)} - w^{(t)}\rangle + \frac{L}{2}\|w^{(t+1)} - w^{(t)}\|^2.$$

As before, we want to bound the two quantities on the right by $\|\nabla f(w^{(t)})\|^2$. Following the bounds from the proof of Theorem 2, provided $\eta^{(t)} < \frac{1}{2\sqrt{d}\|\nabla f_i(w^{(t)})\|}$, we have

$$\nabla f(w^{(t)})^T \sum_{k=2}^{\infty} \frac{1}{k!}\left[\sum_{i_1,i_2\ldots i_k=1}^{d}(-\eta)^k\partial_{l_1,\ldots l_k}\phi_j^{-1}(\phi(w^{(t)}))(\nabla f_{i_t}(w^{(t)})_{l_1}\ldots\nabla f_{i_t}(w^{(t)})_{l_k})\right]$$

$$\leq \frac{\eta^{(t)}\mu}{2\alpha_u L}\|\nabla f(w^{(t)})\|\|\nabla f_{i_t}(w^{(t)})\|.$$

To remove the dependence of $\eta^{(t)}$ on $i_t$, we take $\eta^{(t)} < \frac{1}{2\sqrt{d}\max_i\|\nabla f_i(w^{(t)})\|}$. Since $f$ is $\mu-$PL$^*$ and $f_i$ is non-negative for all $i \in [n]$, $\|\nabla f_i(w^{(t)})\| \leq \sqrt{2Lf_i(w^{(t)})}$. Thus, we can take

$$\eta^{(t)} < \frac{1}{2\sqrt{2dLn}\sqrt{f(w^{(t)})}} \leq \frac{1}{2\sqrt{d}\max_i\|\nabla f_i(w^{(t)})\|}$$

This implies the following bounds:

$$\langle\nabla f(w^{(t)}), w^{(t+1)} - w^{(t)}\rangle \leq -\eta^{(t)}\nabla f(w^{(t)})^T\mathbf{J}_\phi^{-1}(w^{(t)})\nabla f_{i_t}(w^{(t)}) + \left(\frac{\eta^{(t)}\mu}{2\alpha_u L}\right)\|\nabla f(w^{(t)})\|\|\nabla f_{i_t}(w^{(t)})\|,$$

$$\|w^{(t+1)} - w^{(t)}\|^2 \leq \left(\frac{\eta^{(t)2}}{\alpha_l^2} + \frac{\eta^{(t)2}}{4\alpha_u^2}\right)\|\nabla f_{i_t}(w^{(t)})\|^2.$$

Putting the bounds together we obtain:

$$f(w^{(t+1)}) - f(w^{(t)}) \leq -\eta^{(t)} \nabla f(w^{(t)})^T \mathbf{J}_\phi^{-1}(w^{(t)}) \nabla f_{i_t}(w^{(t)}) + \left( \frac{\eta^{(t)}\mu}{2\alpha_u L} \right) \|\nabla f(w^{(t)})\| \|\nabla f_{i_t}(w^{(t)})\|$$

$$+ \left( \frac{\eta^{(t)2}}{\alpha_l^2} + \frac{\eta^{(t)2}}{4\alpha_u^2} \right) \|\nabla f_{i_t}(w^{(t)})\|^2$$

$$\leq -\eta^{(t)} \nabla f(w^{(t)})^T \mathbf{J}_\phi^{-1}(w^{(t)}) \nabla f_{i_t}(w^{(t)}) + \left( \frac{\eta^{(t)}\mu}{2\alpha_u L} \right) 2L \sqrt{f(w^{(t)}) f_{i_t}(w^{(t)})}$$

$$+ \left( \frac{\eta^{(t)2}}{\alpha_l^2} + \frac{\eta^{(t)2}}{4\alpha_u^2} \right) \|\nabla f_{i_t}(w^{(t)})\|^2$$

Now taking expectation over $i_t$, we obtain

$$\mathbb{E}[f(w^{(t+1)})] - f(w^{(t)}) \leq \left( -\frac{\eta^{(t)}}{\alpha_u} \right) \|\nabla f(w^{(t)})\|^2 + \left( \frac{\eta^{(t)}\mu}{\alpha_u} \right) \sqrt{f(w^{(t)})} \mathbb{E}\left[ \sqrt{f_{i_t}(w^{(t)})} \right]$$

$$+ \left( \frac{L\eta^{(t)2}}{2\alpha_l^2} + \frac{L\eta^{(t)2}}{8\alpha_u^2} \right) \mathbb{E}[\|\nabla f_{i_t}(w^{(t)})\|^2]$$

$$\leq \left( -\frac{\eta^{(t)}}{\alpha_u} \right) \|\nabla f(w^{(t)})\|^2 + \left( \frac{\eta^{(t)}\mu}{\alpha_u} \right) f(w^{(t)})$$

$$+ \left( \frac{L\eta^{(t)2}}{2\alpha_l^2} + \frac{L\eta^{(t)2}}{8\alpha_u^2} \right) \mathbb{E}[\|\nabla f_{i_t}(w^{(t)})\|^2]$$

$$\leq \left( -\frac{2\mu\eta^{(t)}}{\alpha_u} \right) f(w^{(t)}) + \left( \frac{\eta^{(t)}\mu}{\alpha_u} \right) f(w^{(t)})$$

$$+ \left( \frac{L\eta^{(t)2}}{2\alpha_l^2} + \frac{L\eta^{(t)2}}{8\alpha_u^2} \right) \mathbb{E}[2L(f_{i_t}(w^{(t)}) - f_{i_t}(w^*))]$$

$$\leq \left( -\frac{\mu\eta^{(t)}}{\alpha_u} + \frac{L^2\eta^{(t)2}}{\alpha_l^2} + \frac{L^2\eta^{(t)2}}{4\alpha_u^2} \right) (f(w^{(t)})).$$

where the second inequality follows from Jensen's inequality and the third inequality follows from Lemma 2. Hence, we have:

$$\mathbb{E}[f(w^{(t+1)})] \leq \left( 1 - \frac{\mu\eta^{(t)}}{\alpha_u} + \frac{L^2\eta^{(t)2}}{\alpha_l^2} + \frac{L^2\eta^{(t)2}}{4\alpha_u^2} \right) (f(w^{(t)})).$$

Now let $C = \left( -\frac{\mu\eta^{(t)}}{\alpha_u} + \frac{L^2\eta^{(t)2}}{\alpha_l^2} + \frac{L^2\eta^{(t)2}}{4\alpha_u^2} \right)$. Then taking expectation with respect to $i_t, i_{t-1}, \ldots i_1$, yields

$$\mathbb{E}_{i_t, \ldots, i_1}[f(w^{(t+1)})] \leq (1 + C)(\mathbb{E}_{i_t, \ldots, i_1}[f(w^{(t)})]$$

$$= (1 + C)(\mathbb{E}_{i_{t-1}, \ldots, i_1}[\mathbb{E}_{i_t | i_{t-1}, \ldots i_1}[f(w^{(t)})]])$$

$$= (1 + C)(\mathbb{E}_{i_{t-1}, \ldots, i_1} f(w^{(t)})]).$$

Hence, we can proceed inductively to conclude that

$$\mathbb{E}_{i_t, \ldots, i_1}[f(w^{(t+1)})] \leq (1 + C)^{t+1}(f(w^{(0)}))).$$

Thus if $0 < 1 + C < 1$, we establish linear convergence. The left hand side is satisfied since $\mu < L$, and the right hand side is satisfied for $\eta^{(t)} < \frac{4\mu\alpha_l^2}{5L^2\alpha_u}$, which holds by the theorem's assumption, thereby completing the proof. $\square$

## G  PROOF OF THEOREM 4

We restate the theorem below.

**Theorem.** *Suppose $\phi : \mathbb{R}^d \to \mathbb{R}^d$ is an invertible, $\alpha_u$-Lipschitz function and that $f : \mathbb{R}^d \to \mathbb{R}$ is non-negative, L-smooth, and $\mu$-PL$^*$ on $\tilde{\mathcal{B}} = \{x \; ; \; \phi(x) \in \mathcal{B}(\phi(w^{(0)}), R)\}$ with $R = \frac{2\sqrt{2L}\sqrt{f(w^{(0)})}\alpha_u^2}{\alpha_l\mu}$. If for all $x, y \in \mathbb{R}^d$ there exists $\alpha_l > 0$ such that*

$$\langle \phi(x) - \phi(y), x - y \rangle \geq \alpha_l \|x - y\|^2,$$

*then,*

(1) *There exists a global minimum $w^{(\infty)} \in \tilde{\mathcal{B}}$.*

(2) *GMD converges linearly to $w^{(\infty)}$ for $\eta = \dfrac{\alpha_l}{L}$.*

(3) *If $w^* = \underset{w \in \tilde{\mathcal{B}} \; ; \; f(w)=0}{\arg\min} \|\phi(w) - \phi(w^{(0)})\|$ then, $\|\phi(w^*) - \phi(w^{(\infty)})\| \leq 2R$.*

*Proof.* The proof follows from the proofs of Lemma 1, Theorem 1, and Theorem 4.2 from Liu et al. (2020). Namely, we will proceed by strong induction. Let $\kappa = \frac{L\alpha_u^2}{\mu\alpha_l^2}$. At timestep 0, we trivially have that $w^{(0)} \in \tilde{\mathcal{B}}$ and $f(w^{(0)}) \leq f(w^{(0)})$. At timestep $t$, we assume that $w^{(0)}, w^{(1)}, \ldots w^{(t)} \in \tilde{\mathcal{B}}$ and that $f(w^{(i)}) \leq (1 - \kappa^{-1})f(w^{(i-1)})$ for $i \in [t]$. Then at timestep $t + 1$, from the proofs of Lemma 1 and Theorem 1, we have:

$$f(w^{(t+1)}) \leq (1 - \kappa^{-1})f(w^{(t)})$$

Next, we need to show that $w^{(t+1)} \in \tilde{\mathcal{B}}$. We have that:

$$
\begin{aligned}
\|\phi(w^{(t+1)}) - \phi(w^{(0)})\| &= \left\| \sum_{i=0}^{t} -\eta \nabla f(w^{(i)}) \right\| \\
&\leq \eta \sum_{i=0}^{t} \|\nabla f(w^{(i)})\| \quad \text{By the Triangle Inequality} \\
&\leq \eta \sqrt{2\frac{L\alpha_u^2}{\alpha_l^2}} \sum_{i=0}^{t} \sqrt{f(w^{(t)}) - f(w^{(t+1)})} \quad (7) \\
&\leq \eta \sqrt{2\frac{L\alpha_u^2}{\alpha_l^2}} \sum_{i=0}^{t} \sqrt{f(w^{(t)})} \\
&\leq \eta \sqrt{2L}\frac{\alpha_u}{\alpha_l} \sum_{i=0}^{t} \sqrt{(1 - \kappa^{-1})^i}\sqrt{f(w^{(0)})} \\
&= \eta \sqrt{2Lf(w^{(0)})}\frac{\alpha_u}{\alpha_l} \sum_{i=0}^{t} (1 - \kappa^{-1})^{\frac{i}{2}} \\
&\leq \eta \sqrt{2Lf(w^{(0)})}\frac{\alpha_u}{\alpha_l} \frac{1}{1 - \sqrt{1 - \kappa^{-1}}} \\
&\leq \eta \sqrt{2Lf(w^{(0)})}\frac{\alpha_u}{\alpha_l} \frac{2}{\kappa^{-1}} \\
&= \frac{\alpha_l}{L} \sqrt{2Lf(w^{(0)})}\frac{\alpha_u}{\alpha_l} 2\frac{\alpha_u L}{\alpha_l\mu} \\
&= \frac{2\sqrt{2L}\sqrt{f(w^{(0)})}\alpha_u^2}{\alpha_l\mu} = R
\end{aligned}
$$

The identity in (7) follows from the proof of $f(w^{(t+1)}) \leq (1 - \kappa^{-1})f(w^{(t)})$. Namely,

$$f(w^{(t+1)}) - f(w^{(t)}) \leq -\frac{L}{2\alpha_u^2}\|-\eta\nabla f(w^{(t)})\|^2$$

$$\implies \|\nabla f(w^{(t)})\| \leq \sqrt{\frac{2\alpha_u^2}{L}}\sqrt{f(w^{(t)}) - f(w^{(t+1)})}$$

$$\implies \|\nabla f(w^{(t)})\| \leq \eta\sqrt{\frac{2L\alpha_u^2}{\alpha_l^2}}\sqrt{f(w^{(t)}) - f(w^{(t+1)})}$$

Hence we conclude that $w^{(t+1)} \in \tilde{\mathcal{B}}$ and so induction is complete. $\qquad\square$

In the case that $\phi^{(t)}$ is time-dependent, we establish a similar convergence result by assuming that $\left\|\sum_{i=1}^{\infty}\phi^{(i)}(w^{(i)}) - \phi^{(i-1)}(w^{(i)})\right\| = \delta < \infty$. Additionally if $\alpha_u^{(t)}$ has a uniform upper bound and $\alpha_l^{(t)}$ has a uniform lower bound, then:

$$\|\phi^{(t)}(w^{(t+1)}) - \phi^{(0)}(w^{(0)})\| = \|\phi^{(t)}(w^{(t+1)}) - \phi^{(t)}(w^{(t)}) + \phi^{(t)}(w^{(t)}) - \phi^{(t-1)}(w^{(t)})$$
$$+ \phi^{(t-1)}(w^{(t)}) - \phi^{(t-1)}(w^{(t-1)}) + \ldots \phi^{(0)}(w^{(1)}) - \phi^{(0)}(w^{(0)})\|$$
$$\leq \left\|\sum_{i=0}^{t}\phi^{(i)}(w^{(i+1)}) - \phi^{(i)}(w^{(i)})\right\| + \left\|\sum_{i=1}^{t}\phi^{(i)}(w^{(i)}) - \phi^{(i-1)}(w^{(i)})\right\|$$
$$\leq R + \delta$$

Hence we would conclude that $\phi^{(t)}(w^{(t+1)}) \in \mathcal{B}(\phi^{(0)}(w^{(0)}), R + \delta)$.

## H   PROOF OF COROLLARY 1 AND COROLLARY 2

We repeat Corollary 1 below.

**Corollary.** *Let $f : \mathbb{R}^d \to \mathbb{R}$ be an L-smooth function that is $\mu$-PL. Let $\alpha_l^{(t)^2} = \min_{i \in [d]}\mathcal{G}_{i,i}^{(t)}$ and $\alpha_u^{(t)^2} = \max_{i \in [d]}\mathcal{G}_{i,i}^{(t)}$. If $\lim_{t \to \infty}\frac{\alpha_l^{(t)}}{\alpha_u^{(t)}} \neq 0$, then Adagrad converges linearly for adaptive step size $\eta^{(t)} = \frac{\alpha_l^{(t)}}{L}$.*

*Proof.* By definition of $\mathcal{G}^{(t)}$, we have that:

$$(1)\ \ \alpha_l^{(t)^2} = \min_{i \in [d]}\mathcal{G}_{i,i}^{(t)}$$

$$(2)\ \ \alpha_u^{(t)^2} = \max_{i \in [d]}\mathcal{G}_{i,i}^{(t)}$$

From the proof of Theorem 1, using learning rate $\eta^{(t)} = \frac{\alpha_l^{(t)}}{L}$ at timestep $t$ gives:

$$f(w^{(t+1)}) - f(w^*) \leq \left(1 - \frac{\mu\alpha_l^{(t)^2}}{L\alpha_u^{(t)^2}}\right)(f(w^{(t)}) - f(w^*))$$

Let $\kappa^{(t)} = \frac{\mu\alpha_l^{(t)^2}}{L\alpha_u^{(t)^2}}$. Although we have that $(1 - \kappa^{(t)}) < 1$ for all $t$, we need to ensure that $\prod_{i=0}^{\infty}(1 - \kappa^{(i)}) = 0$ (otherwise we would not get convergence to a global minimum). Using the assumption that $\lim_{t \to \infty}\frac{\alpha_l^{(t)}}{\alpha_u^{(t)}} \neq 0$, let $\lim_{t \to \infty}(1 - \kappa^{(t)}) = 1 - c < 1$. Then using the definition of the limit, for $0 < \epsilon < c$, there exists $N$ such that for $t > N$, $|\kappa^{(t)} - c| < \epsilon$. Hence, letting

$c^* = \min\left(c - \epsilon, \min_{t \in \{0,1,\ldots N\}} \kappa^{(t)}\right)$, implies that $(1 - \kappa^{(t)}) < 1 - c^*$ for all timesteps $t$. Thus, we have that:

$$\prod_{i=0}^{\infty}(1 - \kappa^{(i)}) < \prod_{i=0}^{\infty}(1 - c^*) = 0$$

Thus, Adagrad converges linearly to a global minimum. $\square$

We present Corollary 2 below.

**Corollary 2.** *Let $f : \mathbb{R}^d \to \mathbb{R}$ be an L-smooth function that is $\mu$-PL. Let $\alpha_l^{(t)^2} = \min_{i \in [d]} \mathcal{G}_{i,i}^{(t)}$. Then Adagrad converges linearly for adaptive step size $\eta^{(t)} = \frac{\alpha_l^{(t)}}{L}$ or fixed step size $\eta = \frac{\alpha_l^{(0)}}{L}$ if $\frac{\alpha_l^{(0)2}}{2L(f(w^{(0)}) - f(w^*))} > \frac{L}{\mu}$.*

*Proof.* By definition of $\mathcal{G}^{(t)}$, we have that:

$$(1) \ \alpha_l^{(t)^2} = \min_{i \in [d]} \mathcal{G}_{i,i}^{(t)}$$

$$(2) \ \alpha_u^{(t)^2} = \max_{i \in [d]} \mathcal{G}_{i,i}^{(t)}$$

In particular, we can choose $\alpha_l = \alpha_l^{(0)}$ uniformly. We need to now ensure that $\alpha_u^{(t)}$ does not diverge. We prove this by using strong induction to show that $\alpha_u^{(t)^2} \leq S$ uniformly for some $S > 0$. The base case holds by Lemma 2 since we have:

$$\alpha_u^{(0)^2} \leq \|\nabla f(w^{(0)})\|^2 = S$$

Now assume that $\alpha_u^{(i)^2} < S$ for $i \in \{0, 1, \ldots t - 1\}$. Then we have:

$$\alpha_u^{(t)^2} \leq \sum_{i=0}^{t} \|\nabla f(w^{(i)})\|^2$$

$$\leq \sum_{i=0}^{t} 2L(f(w^{(i)}) - f(w^*)) \text{ by Lemma 2}$$

$$\leq 2L(f(w^{(0)}) - f(w^*)) \sum_{i=0}^{t-1} \prod_{j=0}^{i} \left(1 - \frac{\mu \alpha_l^{(j)^2}}{L \alpha_u^{(j)^2}}\right)$$

$$\leq 2L(f(w^{(0)}) - f(w^*)) \sum_{i=0}^{t-1} \prod_{j=0}^{i} \left(1 - \frac{\mu \alpha_l^{(0)^2}}{LS}\right)$$

$$\leq 2L(f(w^{(0)}) - f(w^*)) \frac{1}{1 - 1 + \frac{\mu \alpha_l^{(0)^2}}{LS}}$$

$$= 2L(f(w^{(0)}) - f(w^*)) \frac{LS}{\mu \alpha_l^{(0)^2}} < S \text{ by assumption}$$

Hence, by induction, $\alpha_u^{(t)}$ is bounded uniformly for all timesteps $t$.

$\square$

## I    PROOF OF COROLLARY 3

We present the corollary below.

**Corollary 3.** *Suppose $\psi$ is an $\alpha_l$-strongly convex function and that $\nabla\psi$ is $\alpha_u$-Lipschitz. Let $D_\psi(x,y) = \psi(x) - \psi(y) - \nabla\psi(y)^T(x-y)$ denote the Bregman divergence for $x, y \in \mathbb{R}^d$. If $f : \mathbb{R}^d \to \mathbb{R}$ is non-negative, L-smooth, and $\mu$-PL$^*$ on $\tilde{\mathcal{B}} = \{x \; ; \; \nabla\psi(x) \in \mathcal{B}(\nabla\psi(w^{(0)}), R)\}$ with $R = \frac{2\sqrt{2L}\sqrt{f(w^{(0)})}\alpha_u^2}{\alpha_l\mu}$, then:*

(1) *There exists a global minimum $w^{(\infty)} \in \tilde{\mathcal{B}}$ such that $D_\psi(w^{(\infty)}, w^{(0)}) \leq \dfrac{R^2}{2\alpha_l}$.*

(2) *Mirror descent with potential $\psi$ converges linearly to $w^{(\infty)}$ for $\eta = \dfrac{\alpha_l}{L}$.*

(3) *If $w^* = \underset{\{w \; ; \; f(w)=0\}}{\arg\min} D_\psi(w, w^{(0)})$, then $D(w^*, w^{(\infty)}) \leq \dfrac{\alpha_u R^2}{\alpha_l^3} + \dfrac{R^2}{\alpha_l}$.*

*Proof.* The proof of existence and linear convergence follow immediately from Theorem 4. All that remains is to show that $D_\psi(w^{(\infty)}, w^{(0)}) \leq \frac{R^2}{2\mu}$. As $\psi$ is $\alpha_l$-strongly convex, we have:

$$\psi(w^{(\infty)}) \leq \psi(w^{(0)}) + \langle \nabla\psi(w^{(0)}), w^{(\infty)} - w^{(0)}\rangle + \frac{1}{2\alpha_l}\|\nabla\psi(w^{(\infty)}) - \nabla\psi(w^{(0)})\|^2 \quad \text{By Lemma 5}$$

$$\implies D_\psi(w^{(\infty)}, w^{(0)}) \leq \frac{1}{2\alpha_l}\|\nabla\psi(w^{(\infty)}) - \nabla\psi(w^{(0)})\|^2 \leq \frac{R^2}{2\alpha_l}$$

Now let $w^* = \arg\min_{\{w \; ; \; f(w)=0\}} D_\psi(w, w^{(0)})$. Hence $D_\psi(w^*, w^{(0)}) < \frac{R^2}{2\alpha_l}$ by definition. Then we have:

$$\begin{aligned}
D_\psi(w^*, w^{(\infty)}) &\leq \frac{1}{2\alpha_l}\|\nabla\psi(w^*) - \nabla\psi(w^{(\infty)})\|^2 \\
&\leq \frac{1}{2\alpha_l}(2\|\nabla\psi(w^*) - \nabla\psi(w^{(0)})\|^2 + 2\|\nabla\psi(w^{(0)}) - \nabla\psi(w^{(\infty)})\|^2) \\
&\leq \frac{\alpha_u}{\alpha_l}\|w^* - w^{(0)}\|^2 + \frac{R^2}{\alpha_l} \\
&\leq \frac{\alpha_u}{\alpha_l}\frac{2}{\alpha_l}D_\psi(w^*, w^{(0)}) + \frac{R^2}{\alpha_l} \quad \text{By Definition 3} \\
&\leq \frac{\alpha_u R^2}{\alpha_l^3} + \frac{R^2}{\alpha_l}
\end{aligned}$$

$\square$

## J    EXPERIMENTS ON OVER-PARAMETERIZED NEURAL NETWORKS

Below, we present experiments in which we apply the learning rate given by Corollary 1 to over-parameterized neural networks. Since the main difficulty is estimating the parameter $L$ in neural networks, we instead provide a crude approximation for $L$ by setting $L^{(t)} = .99\frac{\|\nabla f(w^{(t)})\|^2}{2f(w^{(t)})}$. The intuition for this approximation comes from Lemma 2. While there are no guarantees that this approximation yields linear convergence according to our theory, Figure 2 suggests empirically that this approximation provides convergence. Moreover, this approximation allows us to compute our adaptive learning rate in practice.

Code for all experiments is available at:

`https://anonymous.4open.science/r/cef30260-473d-4116-bda1-1debdcc4e00a/`

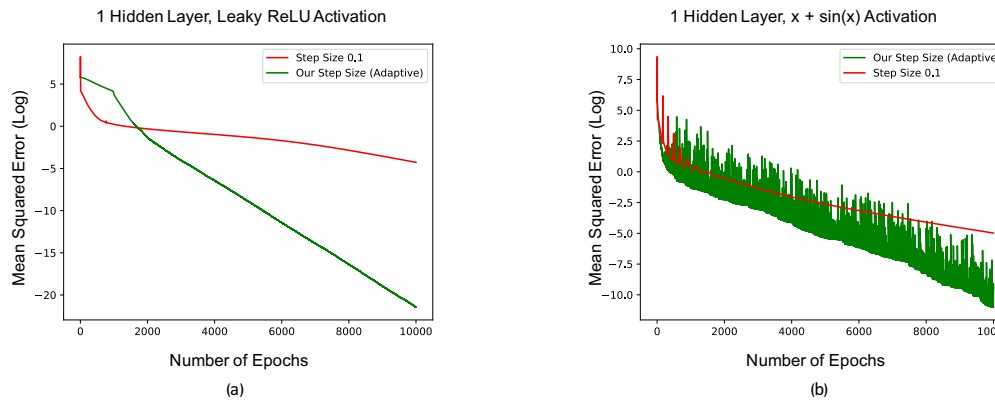

Figure 2: Using the adaptive rate provided by Corollary 1 with $L$ approximated by $L^{(t)} = .99\frac{\|\nabla f(w^{(t)})\|^2}{2f(w^{(t)})}$ leads to convergence for Adagrad in the noisy linear regression setting (60 examples in 50 dimensions with uniform noise applied to the labels). (a) 1 hidden layer network with Leaky ReLU activation Xu et al. (2015) and 100 hidden units. (b) 1 hidden layer network with $x + \sin(x)$ activation with 100 hidden units. All networks were trained using a single Titan Xp, but can be trained on a laptop as well.

