# OpenReview forum: "Linear Convergence and Implicit Regularization of Generalized Mirror Descent with Time-Dependent Mirrors"
_ICLR.cc/2021/Conference — Reject_

### Official Review · AnonReviewer3 · 2020-10-28
**Review of Paper2008**

**Rating:** 5
**Confidence:** 4

**Review:**

This paper studied an algorithm for solving unconstrained smooth finite-sum optimization, called stochastic generalized mirror descent (SGMD). The algorithm SGMD is a generalization of several existing, popular algorithms, including stochastic gradient descent, mirror descent and Adagrad.

The main contribution lies in the convergence rate analysis of the algorithm SGMD based on the Polyak-Lojasiewicz (PL) inequality, which in turn yields linear convergence rate results for some existing methods such as Adagrad. Specifically, the author(s) showed in Theorem 3 that if the objective function satisfies the PL inequality and has a Lipschitz gradient and if the potential function (or called the mirror function) satisfies certain technical assumptions, then SGMD converges linearly to a global minimum. If the PL inequality is satisfied only locally, then local linear convergence result for the GMP (the deterministic version) was also proved. As another contribution, the paper showed that the GMD exhibits an implicit regularization phenomenon in the sense that it converges to a particular optimizers among others.

The Taylor-series-based analysis for stochastic algorithms seems to be new and deserves some merits. However, I do have some doubts about the main results.

First, I'm not sure if one can obtain new, useful algorithms from the general algorithmic framework in the paper and/or deduce new convergence rate results for existing algorithms. If yes, the paper should point it out explicitly, discuss such consequences and compare with the related algorithms/theoretical results. These are not clear from the current presentation of the paper.

The practical implication of the implicit regularization result is also not clear. More efforts should be spent on discussing the meaning or interpretation of the interpolating optimal solution SGMD prioritizes, especially in the context of machine learning problems (e.g., when the optimization problem is the training of neural network or some supervised learning tasks).

---

> ### Author Response · Authors · 2020-11-19
> **Response to Reviewer 3**
>
> We thank the reviewer for the review and address the concerns below.
>
> * “First, I'm not sure if one can obtain new, useful algorithms from the general algorithmic framework in the paper and/or deduce new convergence rate results for existing algorithms. If yes, the paper should point it out explicitly, discuss such consequences and compare with the related algorithms/theoretical results.”
>     * Obtaining new algorithms from our framework is an interesting direction of future work, but is outside the scope of our current paper.  The main novelty of our work is providing explicit learning rates and establishing linear convergence for a large class of optimization methods including mirror descent and adagrad.  The linear convergence rate for mirror descent and Adagrad is novel to the best of our knowledge (the closest related work for Adagrad is the convergence of Adagrad-norm mentioned in the related works section).
> * “The practical implication of the implicit regularization result is also not clear. More efforts should be spent on discussing the meaning or interpretation of the interpolating optimal solution SGMD prioritizes, especially in the context of machine learning problems (e.g., when the optimization problem is the training of neural network or some supervised learning tasks).”
>     * The practical implications of the implicit regularization results (such as generalization, robustness to noise, etc.) are important but are outside the scope of our work.  Our work is primarily focused on establishing linear and local convergence results for a large class of optimization methods.  Properties of solutions for specific mirrors are discussed briefly in the experiments in http://www.its.caltech.edu/~nazizanr/papers/SMD_deep.pdf for reference, but this is a direction of future work.

---

### Official Review · AnonReviewer1 · 2020-10-29
**This line of research is interesting, but I have some concerns.**

**Rating:** 4
**Confidence:** 4

**Review:**

[Summary]
This paper studies the interesting property of generalized mirror descent (GMD) and its stochastic variant for nonconvex optimization problems. First, for GMD this paper shows the linear convergence under PL* condition (in Lemma 1) and finds out a new sufficient condition for the linear convergence (in Theorem 2). Next, this work tried to extend this result to a stochastic setting (in Theorem 3). Moreover, the implicit regularization of GMD is studied, which is an extension of the previous studies by [Azizan et al.].

[Strength]
This paper is easy to read. I think this line of research is very interesting and important to explain the reason why the overparameterized models work well.

[Weakness]
A major concern is the correctness of the statement. There seems to be a technical flaw in the proof of Theorem 3 which is one of main contributions. Indeed, since learning rates $\eta^{(t)}$ depend on $\|f_{i_t}(w^{(t)})\|$, $\eta^{(t)}$ should be also random variable depending on $i_t$. Hence, the equation $E_{i_t}[ \eta^{(t)} \|\nabla f_{i_t}(w^{(t)})\|] =\eta^{(t)} E_{i_t}[\|\nabla f_{i_t}(w^{(t)})\|]$ is invalid. However, this equation is used in the first expression in page 16 in supplementary.
Moreover, it would be better to specify the explicit convergence rate in the main theorems to verify certain linear convergence.

Another concern is the significance of Theorem 4. The property of implicit regularization of GMD is stated in Theorem 4, but this theorem seems not to be a proper extension of the result obtained in [Azizan et al. (2019)].
- The orders wrt R of the ball $\tilde{\mathcal{B}}$ and $\| \phi(w^*)- \phi(w^{(\infty)})\|$ are the same, hence, the convergence of the order $o(R)$ as shown in [Azizan et al. (2019)] does not hold.
- The domain of $w^*$ is restricted in $\tilde{\mathcal{B}}$, but there is no such restriction in [Azizan et al. (2019)].
- If we consider $\phi(w)=w$ which corresponds to the gradient descent, the result (3) is obvious from the triangle inequality. Hence, I am concerned about the importance of this result.

[Improvement]
It would be nice if the authors could mention the above weakness.

---

> ### Author Response · Authors · 2020-11-19
> **Response to Reviewer 1**
>
> We thank the reviewer for their detailed review.  We have updated our submission to address the concerns below.
>
> * “A major concern is the correctness of the statement. There seems to be a technical flaw in the proof of Theorem 3 which is one of main contributions. [...] Moreover, it would be better to specify the explicit convergence rate in the main theorems to verify certain linear convergence.”
>     * We have updated the proof of Theorem 3 to clarify how $\eta^{(t)}$ need not depend on i_t, and now provide a constant step size for convergence when the function \phi is not time-dependent.  We chose to keep the statements of the theorems simple by not including the explicit coefficients for linear convergence.  We do provide all of these in the appendix and have updated the manuscript to present an example in the remark after Theorem 1.  Note that since the mirror is time-varying, we would have a time varying coefficient for linear convergence.  However, these coefficients are all bounded away from 1 by assumption and so we get linear convergence.
>
>
> Comparison to Azizan et al. 2019:
>
> * "The orders wrt R of the ball $\tilde{\mathcal{B}}$ and $||\phi(w^*) - \phi(w^{(\infty)}||$ are the same, hence, the convergence of the order $o(\epsilon)$ as shown in [Azizan et al. (2019)] does not hold."
>     * We thank the reviewer for pointing this out, and have updated the manuscript to clarify our point further. Namely, our Theorem 4 and Corollary 3 imply Assumption 1 used in Theorem 3 & 4 from Azizan et al. 2019 by proving that there exists an interpolating solution within a ball in the dual space and provides conditions under which generalized mirror descent converges linearly to this solution.
> * “The domain of $w^*$ is restricted in $\tilde{\mathcal{B}}$, but there is no such restriction in [Azizan et al. (2019)].”
>     * In both our paper and Azizan et al. (2019, by definition, the point $w^*$ needs to be inside the corresponding ball.  Azizan et al. (2019) assumes that there is an interpolating solution within a ball in Bregman divergence space and thus the nearest interpolating solution must also be within this ball.  We do not restrict the domain of w^* to be in $\tilde{\mathcal{B}}$, but rather, we first prove that there exists an interpolating solution within this ball and hence, there is naturally a closest interpolating solution in this ball.  This is a key distinction between our work and Azizan et al. 2019 Assumption 1, which assumes that there exists an interpolating solution within a ball B in the Bregman divergence space.
> * “the result (3) is obvious from the triangle inequality. Hence, I am concerned about the importance of this result.”
>     * This result follows from the triangle inequality, but the main point of our theorem is that we prove that there is an interpolating solution within the ball instead of assuming this.

---

### Official Review · AnonReviewer4 · 2020-10-30
**Needs further clarifications**

**Rating:** 5
**Confidence:** 4

**Review:**

The paper studies 1) the convergence rate; and 2) the implicit regularization of (stochastic)GMD, a generalization of (stochastic) mirror descent where mirror maps can be time dependent. The contributions are as follows: 1) Theorem 1: for PL+smooth functions, the paper presents a linear convergence result for GMD; 2) Theorem 2, 3: they provide sufficient conditions on the Jacobian of the mirror map, under which (stochastic)GMD converges linearly; 3) Theorem 4: argues for an “approximate implicit regularization” effect of GMD, by showing that GMD converges to a point, “close” to the initialization, where closeness is measured in terms of the \ell_2 distance in the mirrored space.

- Correctness: I have a problem understanding the proof of Theorem 3. In the fifth line of the proof of theorem 3, page 15 of the appendix, I’m not sure how you upper bound the quantity - < \nabla f(w^(t)), J_\phi^-1 \nabla f_{i_t}(w^(t)) > by the right hand side. Of course, the Jacobian is assumed to be positive definite, but how does this upper bound follow? Besides, I found the claim in Theorem 3 quite interesting, and even surprising. It seems to me that SGD, which is SGMD with mirror maps equal to identity, satisfies the conditions of Theorem 3, simply because the mirror map is the identity. Then, the result claims an exponential convergence rate for SGD to the global optima. Can you clarify this, in light of the minimax lower bounds for SGD in smooth and strongly convex setting, e.g. the recent work: https://papers.nips.cc/paper/8624-tight-dimension-independent-lower-bound-on-the-expected-convergence-rate-for-diminishing-step-sizes-in-sgd.pdf

- Novelty/Significance of the results: Theorem 3 is novel and seems very interesting, if the authors clarify the issue above, as well as clarify the generality of the assumption, i.e. the conditions required for the Jacobian of the mirror map. Can you clarify what are the technical challenges in proving Theorem 1, and what are the novelties with respect to several related work, including the work of Karimi et al., 2016?

- Clarity: In my humble opinion, Theorem 4 cannot be interpreted as an “approximate implicit regularization” effect, simply because the distance to the optimum measured in the mirrored space can be very large, and the authors do not provide examples when this can actually be small. In the remarks, you point out “Hence provided that R is small (which holds for small f(w(0))), GMD selects an interpolating solution that is close to w*  in \ell_2-norm in the dual space”. But why should one expect f(.) to be small at the initialization? Specifically, this theorem is proved under the assumption that f(.) is non-negative, therefore, f(w^(0)) being small simply means that w^(0) is already an approximate global optimum.

- Presentation: I think there is room for improving the writeup. Here are some examples:
   + The presentation of MD-type updates in Equation (1) is quite abrupt, without providing the motivation/context.
   + All of the main theorems guarantee linear convergence without actually characterizing the rate of convergence. I found it very confusing as it hides the dependence on important parameters.
   + The presentation of the PL* condition in section 3 is very confusing as none of the main results (Theorems 1-3) requires this condition. What is the motivation? It seems that this condition is not preserved under simple transformations such as translation, i.e. if f(X) satisfy PL* then f(x) + 1 will not. What are some interesting function classes that satisfy this property?
   + Do the assumptions of Lemma1 (PL*, smoothness, non-negativity) imply existence of x* such that f(x*) = 0?
   + Can the monotonicity condition in Theorem 4 be restated in terms of strong/strict convexity of the potential function associated with the mirror map?
   + The sufficient conditions in theorems 2 and 3 are not discussed at all. What are some examples of interesting potential functions / Bregman divergences that satisfy these conditions?

For the reasons listed above, I don’t think this paper is ready to be published at ICLR. Of course, I will happily reconsider my evaluation and increase my score if the authors clarify the issues raised above.

****************************************Post-rebuttal comments****************************************
***************************************************************************************************
After reading authors feedback, I've increased my score from 3 to 5; however, the paper in its current form is still a borderline.
***************************************************************************************************

---

> ### Author Response · Authors · 2020-11-19
> **Response to Reviewer 4**
>
> We thank the reviewer for their detailed review.  We have updated our submission to reflect these changes and address the concerns below.
>
> * “I have a problem understanding the proof of Theorem 3. In the fifth line of the proof of theorem 3, page 15 of the appendix,”
>     * Thank you for pointing this out.  We have corrected the proof to resolve this (the changes are reflected in our updated submission).  The change was just that we do not upper bound this term until after taking the expectation with respect to sample i_t.   In fact, we have now strengthened the results to establish convergence under a fixed learning rate as well.
> * “ Can you clarify this, in light of the minimax lower bounds for SGD in smooth and strongly convex setting, e.g. the recent work”
>     * Regarding the minimax lower bound for SGD, when the mirror map is the identity, our results rely on over-parameterization.  This is different from the setting of the paper establishing the minimax lower bounds for SGD, and our result is indeed consistent with the result from https://arxiv.org/pdf/1811.02564.pdf for SGD in the over-parameterized setting.  Namely, our Theorem 3 relies on the assumption that our losses PL* and non-negative, and thus our convergence result implies the existence of an interpolating solution.
>  * “Theorem 3 is novel and seems very interesting, if the authors clarify the issue above, as well as clarify the generality of the assumption, i.e. the conditions required for the Jacobian of the mirror map.”
>     * Thank you again for finding our results novel and interesting.  The assumptions of Theorem 3 are mild since they cover positive definite preconditioner methods.  We also note that while we have required bounded derivatives for \phi everywhere to simplify our statement, we really only require these conditions in a ball that contains the optimization path.
> * “Can you clarify what are the technical challenges in proving Theorem 1, and what are the novelties with respect to several related work, including the work of Karimi et al., 2016?”
>     * The main technical challenge in proving Theorem 1 is that the updates are in the dual space (i.e. after transforming by \phi), and so we needed to identify appropriate conditions on \phi under which we could apply a PL-based proof. Hence, one key novelty of our work is demonstrating that a PL-based analysis is possible for generalized mirror descent.  While the work by Karimi et al. 2016 provides a PL-based analysis for gradient descent, we identified additional conditions on \phi under which this analysis could be extended to generalized mirror descent.  Theorem 1 is especially interesting since for mirror descent, these conditions essentially reduce down to the potential function being strongly convex (as discussed in Section 4).  This condition is assumed in other work on understanding the convergence of mirror descent (see corollary 7 of http://www.its.caltech.edu/~nazizanr/papers/SMD.pdf), but has not been connected to linear convergence before.
> * “In my humble opinion, Theorem 4 cannot be interpreted as an “approximate implicit regularization” effect, [...] But why should one expect f(.) to be small at the initialization? Specifically, this theorem is proved under the assumption that f(.) is non-negative, therefore, f(w^(0)) being small simply means that w^(0) is already an approximate global optimum.”
>     * We agree that there is apriori no reason to assume that f(w^0) is small.  However, our Theorem 4 and Corollary 3 imply Assumption 1 of Azizan et al. (http://www.its.caltech.edu/~nazizanr/papers/SMD_deep.pdf) and establish that when f(w^0) is small, mirror descent converges to a solution that is close to the nearest interpolating solution in the dual space.  A similar case is analyzed in http://www.its.caltech.edu/~nazizanr/papers/SMD_deep.pdf.

---

> > ### Author Response · Authors · 2020-11-19
> > **Response Part 2**
> >
> > Regarding the comments on presentation:
> >
> > * “The presentation of MD-type updates in Equation (1) is quite abrupt, without providing the motivation/context.”
> >     * We would be happy to add more discussion around the introduction of MD type updates in Equation 1.
> > * “All of the main theorems guarantee linear convergence without actually characterizing the rate of convergence. I found it very confusing as it hides the dependence on important parameters.”
> >     * We chose to keep the statements of the theorems simple by not including the explicit coefficients for linear convergence.  We do provide all of these in the appendix and have updated the manuscript to include a simple case in the remark after Theorem 1.  Note that since the mirror is time-varying, we would have a time varying coefficient for linear convergence.  However, these coefficients are all bounded away from 1 by assumption and so we get linear convergence.
> > * “The presentation of the PL* condition in section 3 is very confusing as none of the main results (Theorems 1-3) requires this condition. [...] Do the assumptions of Lemma1 (PL*, smoothness, non-negativity) imply existence of x* such that f(x*) = 0?”
> >     * The PL* condition is actually used in Theorem 3.  We have updated the manuscript to reflect this change.  The main motivation is that for non-negative functions, our assumptions plus the PL* condition imply the existence of x* such that f(x*) = 0 (i.e. x* is an interpolating solution).
> > * “Can the monotonicity condition in Theorem 4 be restated in terms of strong/strict convexity of the potential function associated with the mirror map?”
> >     * Just to make sure we understand the question about monotonicity - are you referring to the condition on the inner product for \phi being lower bounded by \alpha_l \| x -y \|^2 ?  If so, then this can be restated in terms of strong convexity of the potential function when \phi is the gradient of a potential function (as in Section 5).  However, we avoid doing so since we can keep full generality in our results even when \phi is not the gradient of a potential.
> > * “What are some examples of interesting potential functions / Bregman divergences that satisfy these conditions?”
> > Understanding which functions in the mirror descent setting satisfy the conditions of Theorem 2 and 3 is an interesting future work.  However, we note that pre-conditioner methods do fall in this setting since the higher order derivatives are 0.

---

### Official Review · AnonReviewer2 · 2020-11-03
**Analyses are standard and very restrictive**

**Rating:** 3
**Confidence:** 4

**Review:**

##########################################################################


Summary:

This paper shows a “generalized mirror descent“ converges linearly if the objective function is smooth and satisfies the Polyak-Lojaciewicz condition.


##########################################################################


Reasons for score:


The novelty seems to be very limited. This paper does not actually show the benefit of replacing the gradient of the Bregman function by a general mapping in mirror descent. The analyses are standard and do not actually get more difficult with the generalization. The theorems require the derivatives of the Bregman function to be bounded, making the result not even applicable to entropic mirror descent, arguably the most notable instance of mirror descent.


##########################################################################Pros:


Pros:

1. None. I have not seen a proof showing mirror descent converges linearly with the Polyak-Lojaciewicz condition in literature, so this could be a novelty. But the analyses in this paper are just standard and very restrictive because they require the Bregman functions to have bounded derivatives.


##########################################################################

Cons:

1. I do not see the benefit of replacing the gradient of the Bregman function by a general mapping \phi. The authors use gradient descent, mirror descent, and AdaGrad as examples of the “generalized mirror descent” in Section 5. However, it is already known that gradient descent is a special case of mirror descent and AdaGrad is mirror descent with a time-varying Bregman function. All three do not need the notion of a “generalized mirror descent.”

2. The proposed “generalized mirror descent” is not actually more general than mirror descent. The assumption on \phi in Theorem 1, for example, coincides with the standard assumption in mirror descent literature that the Bregman divergence is strongly convex. Moreover, the condition that \phi is Lipschitz is restrictive (for example, it does not hold for entropic mirror descent) and does not appear in standard mirror descent literature. The conditions on the derives of \phi are even more restrictive in Theorem 2 and Theorem 3.

3. The theorems in this paper only states that the algorithms converge linearly. Please make an optimal choice of the step sizes and make the dependence of the convergence rate on the problem and algorithm parameters explicit. Also please make sure that the exponent in the linear convergence rate does not depend on the iteration counter, as opposed to those shown in the appendix.

4. The paper starts with a discussion on overparameterization. However, I do not see the connection of this paper with explaining the benefit of overparameterization. Indeed, the analyses in this paper has nothing to do with overparameterization, unless the authors connects the PL condition with overparameterization.

5. Isn’t the PL^* condition simply the PL condition with the objective function shifted by -f(x^*)? As long as f(x^*) is finite, without loss of generality, one can always do such a shifting and the sequence of iterates does not change. I do not think PL^* is a new condition.


##########################################################################

After reading the rebuttal:

I keep the score.

- In my view, the Taylor-series analysis is a very straightforward approach such that existing frameworks apply to the setup considered in this paper. Such a straightforward approach renders the results restrictive and hence I do not consider it a significant novelty.

The proof of Theorem 1 just rewrites the existing proof with minor modifications for the considered setup and hence do not require any additional restriction. Emphasizing Theorem 1 is not very appropriate for the rebuttal.

- The "generalization" I mentioned is for the \phi function. There are already several works on mirror descent/FTRL with a time-varying regularizer (Orabona et al. (2015) is for FTRL, as clarified in Orabona's lecture notes on online learning). Without an example where the \phi function is not the gradient of a potential function, I do not see the necessity of this generalization.

For the same reason above, emphasizing Theorem 1 is not very appropriate for the rebuttal.

- My point is to clarify the dependence of the convergence rate on the problem parameters. I do not think the proofs in the appendices provide explicit characterizations of such dependence. To make such explicit characterizations, I think specifying *a* step size instead of *a range* of step sizes is perhaps necessary.

- I suggest the authors rewrite the first two paragraphs to make the motivation of the problem setup clearer there. Section 2 is OK.


#########################################################################

---

> ### Author Response · Authors · 2020-11-19
> **Response to Reviewer 2**
>
>
> We thank the reviewer for their review, and we address the concerns below:
>
> * “None. I have not seen a proof showing mirror descent converges linearly with the Polyak-Lojaciewicz condition in literature, so this could be a novelty. But the analyses in this paper are just standard and very restrictive because they require the Bregman functions to have bounded derivatives.”
>     * We respectfully disagree that there is no novelty in our work.  The proof that generalized mirror descent converges linearly under the PL inequality is novel and our Theorem 1 does not require bounded derivatives.  Our bound on the derivatives is only used for addressing stochastic generalized mirror descent and our derivative bounds are mild since they involve a factorial term.  We would also like to point out that our Taylor series analysis is non-standard and that our local convergence results are novel as well.
> * “I do not see the benefit of replacing the gradient of the Bregman function by a general mapping \phi. [...], it is already known that gradient descent is a special case of mirror descent and AdaGrad is mirror descent with a time-varying Bregman function. All three do not need the notion of a “generalized mirror descent.” ”
>     * The main benefit of our formulation is that it precisely generalizes and simplifies these special cases.  Hence, we need to only establish our bounds in the general setting for it to apply directly to all these cases.  We additionally note that this generalization of mirror descent has been studied in prior work as well (which we discuss in our related works section): https://arxiv.org/abs/1304.2994.
> * “The proposed “generalized mirror descent” is not actually more general than mirror descent.”
>     * When \phi is the gradient of a potential function and when \phi is not changing over time, then the conditions of theorem 1 do indeed coincide with the standard assumption of strongly convex potential function for mirror descent.  However, this reinforces that the standard setting of mirror descent with strongly convex potential function is a special case of generalized mirror descent.  Moreover, generalized mirror descent has already been introduced as a generalization of mirror descent in prior work  https://arxiv.org/abs/1304.2994.
> * “The theorems in this paper only states that the algorithms converge linearly. Please make an optimal choice of the step sizes and make the dependence of the convergence rate on the problem and algorithm parameters explicit.”
>     * We provide step size choices for linear convergence in all of our theorems.  To keep the statement of the theorems as simple as possible, the rates of convergence are provided explicitly in the proofs in the appendix.  We have updated our manuscript to provide an example of such a rate in the remark after theorem 1.  Note that since the mirror is changing at every time step, the coefficient in the convergence rate will of course depend on the values of \alpha_u and \alpha_l for the given mirror.  However, provided that these quantities are bounded, we can upper bound coefficient for the rate of convergence to establish linear convergence.  We explain this in the remark after theorem 1, but are happy to clarify this further if needed.
> * “The paper starts with a discussion on overparameterization. However, I do not see the connection of this paper with explaining the benefit of overparameterization.”
>     * The main connection with over-parameterization and the PL and PL* conditions are outlined in our related work.  Namely, we indicate that https://arxiv.org/abs/2003.00307 demonstrate that the PL* condition holds for modern over-parameterized neural networks and thus our results hold for this setting as well.  Moreover, the PL* condition and non-negativity of the loss function imply linear convergence to an interpolating solution, and thus over-parameterization is necessary as it allows for interpolation.
> * “Isn’t the PL^* condition simply the PL condition with the objective function shifted by -f(x^*)? [...] I do not think PL^* is a new condition. “
>     * We do not claim that the PL^* condition is new, and we reference prior work (https://arxiv.org/abs/2003.00307) that uses this condition.  It is indeed a special case of PL. The benefit of the PL^* condition is that for non-negative loss functions that are \mu-PL*, we are able to show that GMD and SGMD converge linearly to an interpolating solution without assuming such a solution exists.

---

### Decision · Program_Chairs · 2021-01-07
**Final Decision**

**Decision:**

Reject

**Comment:**

The paper shows linear convergence for generalized mirror descent on smooth function under the PL assumption. It extends the result to stochastic generalized mirror descent under an additional assumption on the Jacobian of the mirror map. Reviewers pointed out several technical issues with the submission. While some of the problems have since been resolved in the updated version, the paper still lacks sufficient novelty, and some concerns regarding the correctness/clarity of the claims remain. Unfortunately, I can not recommend acceptance at this time.